# CK2-mediated phosphorylation of SUZ12 promotes PRC2 function by stabilizing enzyme active site

Lihu Gong [1,4], Xiuli Liu[1,4], Lianying Jiao[1,3,4], Xin Yang[1], Andrew Lemoff [2] & Xin Liu [1] ✉

Polycomb repressive complex 2 (PRC2) plays a key role in maintaining cell identity during differentiation. Methyltransferase activity of PRC2 on histone H3 lysine 27 is regulated by diverse cellular mechanisms, including post-translational modification. Here, we report a unique phosphorylation-dependent mechanism stimulating PRC2 enzymatic activity. Residue S583 of SUZ12 is phosphorylated by casein kinase 2 (CK2) in cells. A crystal structure captures phosphorylation in action: the flexible phosphorylation-dependent stimulation loop harboring S583 becomes engaged with the catalytic SET domain through a phosphoserine-centered interaction network, stabilizing the enzyme active site and in particular S-adenosyl-methionine (SAM)-binding pocket. CK2-mediated S583 phosphorylation promotes catalysis by enhancing PRC2 binding to SAM and nucleosomal substrates and facilitates reporter gene repression. Loss of S583 phosphorylation impedes PRC2 recruitment and H3K27me3 deposition in pluripotent mESCs and compromises the ability of PRC2 to maintain differentiated cell identity.

Polycomb repressive complex 2 (PRC2) is a key epigenetic enzyme complex involved in the maintenance of cell identity during stem cell differentiation[1,2]. PRC2 catalyzes methylation of histone H3 lysine 27 (H3K27); trimethylated H3K27 (H3K27me3) is a hallmark of gene silencing[3–6]. PRC2 plays roles in both oncogenesis and tumor suppression in a cell context-dependent manner by, for example, conferring transcriptional repression of cell cycle checkpoint genes and proliferation genes, respectively[2,7]. The PRC2 core complex consists of four subunits: EZH2 (or its paralog EZH1) serves as the catalytic subunit; other core subunits include EED, SUZ12, and RBBP4 (or its paralog RBBP7). EZH2, EED, and the C-terminal VEFS (VRN2, EMF2, FIS2, and SU(Z)12) domain of SUZ12 (SUZ12(VEFS)) assemble into the minimally active catalytic module[8,9], whereas RBBP4 and the N-terminal region of SUZ12 are together folded into the accessory subunit-binding module,

which associates with a series of developmentally regulated accessory subunits in PRC2 holo complexes, modulating chromatin binding[10–13].

A focal point of the cellular regulation of PRC2 function is methyltransferase activity. The PRC2 core complex displays limited basal activity. The existing H3K27me3 histone mark engages with the aromatic cage of EED and allosterically stimulates PRC2 enzymatic activity[8,9,14]. PRC2 stimulation by H3K27me3 is thought to at least in part account for the spreading of H3K27me3 on repressive chromatin[14]. For genomic loci devoid of H3K27me3, JARID2 with tri-methylated lysine 116 (JARID2K116me3) can initiate H3K27me3 deposition by activating PRC2 through a similar allosteric mechanism[15]. Local chromatin compaction accompanied by a distinct linker DNA length represents another cellular process leading to PRC2 activation, although the underlying molecular basis is not completely

[1]Cecil H. and Ida Green Center for Reproductive Biology Sciences, University of Texas Southwestern Medical Center, Dallas, TX 75390, USA. [2]Department of Biochemistry, University of Texas Southwestern Medical Center, Dallas, TX 75390, USA. [3]Present address: Department of Biochemistry and Molecular Biology, School of Basic Medical Sciences, Xi'an Jiaotong University Health Science Center, Xi'an, Shaanxi 710061, China. [4]These authors contributed equally: Lihu Gong, Xiuli Liu, Lianying Jiao. ✉e-mail: xin.liu@utsouthwestern.edu

understood[16]. In comparison, Y641F/N/S/H and A677G cancer mutations of EZH2 found in human B-cell lymphomas cause hyper-trimethylation of H3K27 in a heterozygous genetic background by directly remodeling the active site and changing product specificity[17,18]. PRC2 enzymatic activity is also subjected to inhibition by distinct cellular mechanisms. As a notable example, oncogenic H3K27M mutant histone identified in diffuse midline gliomas globally diminishes H3K27me3 level[19], by blocking the histone substrate-binding channel of PRC2 in a SAM-dependent manner[9,20,21]. Interestingly, EZHIP expressed normally in gonads—and abnormally in posterior fossa ependymoma—restricts PRC2 activity with a protein sequence mimicking H3K27M[22–25].

In addition to the methyltransferase activity, the establishment of cell type-specific H3K27me3 patterns depends on accurate chromatin targeting of PRC2. There are two classes of PRC2 holo complexes, PRC2.1 and PRC2.2, in mammalian cells, which colocalize at many target sites in mouse embryonic stem cells (mESCs). PRC2.1 and PRC2.2 are defined based on the types of accessory subunits bound to the core complex: PHF1/MTF2/PHF19 (a.k.a. PCL1/2/3), EPOP, and PALI1/PALI2 are components of PRC2.1, whereas AEBP2 and JARID2 belong to PRC2.2[26–28]. Combined genetic ablation of the accessory subunits from both holo complexes obliterates chromatin enrichment of PRC2 and results in a dispersed H3K27me3 pattern throughout the genome[29,30]. In human-induced pluripotent stem cells (iPSCs), PRC2.1 and PRC2.2 compete for overlapping target sites; these holo complexes correlate with disparate H3K27me3 levels and varying degrees of gene repression, possibly due to differences in chromatin binding affinity[31].

PRC2 subunits undergo extensive posttranslational modification (PTM), such as reversible phosphorylation, which couples cell signaling to PRC2-mediated epigenetic gene silencing[32]. For example, phosphorylation of residue S21 of EZH2 by AKT kinase hampers H3K27 methylation and causes derepression of developmental genes in several cancer cell lines[33]. Cyclin-dependent kinase 1 (CDK1) phosphorylates residue T345 of EZH2, promoting PRC2 recruitment and H3K27me3 deposition at target loci[34,35]. AMP-activated protein kinase (AMPK) is responsible for phosphorylation of residue T311 of EZH2 upon energy deprivation, which suppresses H3K27 trimethylation and inhibits tumor cell growth[36]. Much less is known about posttranslational modification of SUZ12, except that phosphorylation of residues S539, S541, and S546 by polo-like kinase 1 (PLK1) has been found to facilitate proteasomal degradation of PRC2 in liver tumors[37].

CK2 is a conserved, ubiquitously expressed protein kinase, which displays broad substrate specificity[38,39]. Active CK2 in mammalian cells adopts a 2:2 tetrameric structure, containing two catalytic subunits, CK2α/CK2α′, and two regulatory subunits, CK2β[38,39]. CK2 is a component of two variant Polycomb repressive complex 1 (PRC1), PRC1.3 and PRC1.5[28,40]. CK2 inhibits monoubiquitination of histone H2A lysine 119 (H2AK119) mediated by PRC1.5[41]. Notably, monoubiquitinated H2AK119 (H2AK119ub) has recently been shown to play a direct role in the chromatin recruitment of PRC2[42–48]. CK2 expression and activity positively correlate with proliferation and survival of cancer cells, and host cell CK2 is exploited by several viruses, including COVID-19, to promote viral life cycle[38,39,49,50]; inhibition of CK2 enzymatic activity by chemical compounds is being tested in clinical trials for the treatment of coronavirus disease caused by COVID-19 and of various cancer types, including cholangiocarcinoma, basal cell carcinoma (BCC), and recurrent medulloblastoma (clinicaltrials.gov).

Although the catalytic mechanism of PRC2 in both basal and H3K27me3-stimulated states has been subjected to extensive biochemical and structural studies[8,9], our understanding of how the enzyme may be regulated in cells remains far from complete. Here, we report a unique phosphorylation-dependent mechanism that promotes PRC2 function in cells. CK2 mediates SUZ12 phosphorylation at a serine residue located in the SUZ12(VEFS) domain. A crystal structure captures the phosphorylated SUZ12 in action: it induces structural remodeling of an otherwise flexible acidic loop region in the SUZ12(VEFS) domain, establishing a set of molecular interactions with the catalytic SET [Su(var)3–9, Enhancer-of-zeste and Trithorax] domain to stabilize the enzyme active site and in particular SAM-binding pocket. SUZ12 phosphorylation increases PRC2 enzymatic activity, enhances PRC2 binding to nucleosomes, and promotes reporter gene repression. Loss of this phosphorylation in mESCs not only reduces PRC2 enrichment and H3K27me3 deposition, but also impairs the ability of PRC2 to maintain a differentiated state of mESCs.

## Results

### Residue S583 of human SUZ12 is phosphorylated in vivo

Phosphorylation of residue S583 of human SUZ12 (SUZ12S583) and its mouse equivalent mSUZ12S585 has been previously noted in several untargeted phosphoproteomics studies (Fig. 1a)[51]. To confirm this phosphorylation in a targeted low throughput assay, we purified endogenous PRC2 from mESCs using an anti-SUZ12 affinity column and carried out PTM analysis using liquid chromatography-tandem mass spectrometry (LC-MS/MS). MS/MS spectra clearly indicated the presence of phosphorylated mSUZ12S585 (mSUZ12S585p) in vivo (Supplementary Fig. 1). Semi-quantitative assessment based on the abundance of the peptides with or without the PTM indicated that the majority of mSUZ12 is phosphorylated at this site in mESCs (Fig. 1b).

To characterize SUZ12S583 phosphorylation in human cell lines, we raised a rabbit polyclonal anti-SUZ12S583p antibody using a synthetic peptide encompassing SUZ12S583p. The purified antibody displayed at least 32-fold discrimination between phospho- and apo peptides (Fig. 1c). In addition, phospho- but not apo peptide blocked antibody binding to phosphorylated SUZ12 from HEK293T nuclear extracts (Supplementary Fig. 2), indicative of phospho-specific recognition of SUZ12 by the antibody. In line with this result, antibody signals were greatly diminished by either treatment of a five-member PRC2 holo complex (PRC2-5m), EZH2–EED–SUZ12–RBBP4–AEBP2, with λ protein phosphatase or introduction of an S583A mutation on SUZ12 in the same complex (Fig. 1d).

Using the developed antibody, we examined SUZ12S583 phosphorylation in various cancer cell lines in a semi-quantitative manner. We found that SUZ12S583 phosphorylation is a widespread phenomenon (Fig. 1e and Supplementary Fig. 3). Compared to the total cellular SUZ12, the SUZ12S583 phosphorylation level displayed cell line-specific variations, with some cell lines showing distinctly less phosphorylation (Fig. 1e and Supplementary Fig. 3), which may be accounted for by different kinase activities accessible to SUZ12 in these cells.

### CK2 mediates phosphorylation of residue S583 of SUZ12

To identify the kinase responsible for SUZ12S583 phosphorylation in cells, we first performed a search with two web servers, NetPhos 3.1 and PhosphoNET[52] (www.phosphonet.ca), both of which predicted protein kinase CK2 as the top candidate (Fig. 2a). Manual inspection also indicated the existence of a potential CK2 substrate motif based on a compilation of known phosphorylation motifs (Fig. 1a)[53]. To experimentally validate the prediction, we purified two versions of CK2, α2β2 and α′2β2, and carried out in vitro phosphorylation assay on bacterially expressed SUZ12. CK2-α2β2 and CK2-α′2β2 were able to phosphorylate GST-tagged SUZ12 equally well, as indicated by an anti-phosphoserine antibody recognizing all phosphorylated serine residues (Fig. 2b). An S583A mutation nearly abolished phosphorylation, whereas alanine mutation of two other nearby serine residues, S546 and S604, only moderately reduced phosphorylation (Fig. 2b), suggesting S583 is the primary target of CK2 kinase activity on SUZ12.

To study CK2-mediated SUZ12 phosphorylation in vivo, we used shRNAs to knock down the CK2 subunit α, α′, or β in an embryonic carcinoma cell line NT2/D1. Knockdown efficiency of two different

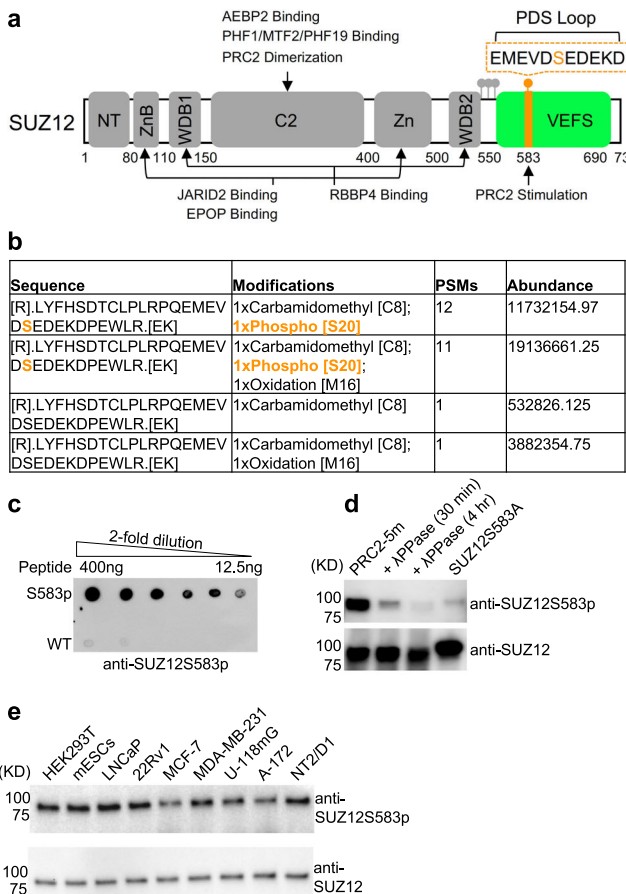

**Fig. 1 | Residue S583 of human SUZ12 is phosphorylated. a** Domain structure of SUZ12. Structurally characterized SUZ12 domains are represented by gray blocks except that the VEFS domain included in the current study is colored in green. The PDS loop harboring S583 is highlighted in orange with the amino acid sequence shown above. **b** Peptides identified for SUZ12 by LC-MS/MS which contain S583. Peptide sequence, modifications, number of peptide spectrum matches (PSMs), and peptide abundance are listed. The phosphorylated residue that was unambiguously assigned is shown in orange. The percentage of phosphorylation was calculated based on a comparison of the abundances of the phosphorylated and unphosphorylated peptides. **c** Dot blot. Apo and phosphorylated peptides were applied on a nitrocellulose membrane with a serial dilution. Phospho-specific reactivity of the developed anti-SUZ12S583p antibody was analyzed. A representative of three independent experiments is shown. **d** Effect of λ phosphatase treatment and serine mutation on ectopically expressed PRC2-5m, EZH2–EED–SUZ12–RBBP4–AEBP2. The total amount of PRC2-5m is indicated by anti-SUZ12 signals. S583 phosphorylation level is indicated by signals of the anti-SUZ12S583p antibody developed in this study (uncropped gel images of this figure are shown in Supplementary Fig. 16). A representative of three independent experiments is shown. **e** Levels of S583 phosphorylation in stem cells and cancer cells. Anti-SUZ12S583p signals were generated using immunoprecipitates of anti-SUZ12 antibody to avoid a non-relevant contaminating band (also see Supplementary Fig. 2). A representative of three independent experiments is shown. Source data are provided as a Source Data file.

shRNAs in each case was confirmed by respective antibodies (Fig. 2c). The SUZ12S583 phosphorylation level was markedly decreased by the loss of the CK2 catalytic subunit α or α′ and, to a larger extent, the shared regulatory β subunit (Fig. 2d). CX4945 (silmitasertib) is a potent and highly selective chemical inhibitor of CK2 that is being clinically tested in anti-cancer and anti-virus therapies. Treatment of HEK293T cells, mESCs, and a panel of cancer cell lines by CX4945 resulted in a dose-dependent diminution of SUZ12S583 phosphorylation (Fig. 2e), further supporting the role of CK2 as the specific kinase for SUZ12S583 phosphorylation in these cells.

## S583 phosphorylation stabilizes enzyme active site

Residue S583 is located in the VEFS domain of SUZ12, which associates with the SET domain of EZH2 and is essential for the enzymatic activity[8,9]. The highly conserved acidic sequence surrounding S583 on SUZ12 was previously implicated in the stimulation of PRC2 enzymatic activity (Fig. 3a)[16,54,55]. However, this acidic loop region is not well defined in the known structures of 2.6–3.0 Å resolution in the absence of S583 phosphorylation (Supplementary Fig. 4a, b)[9,56], making it difficult to predict the impact of S583 phosphorylation on PRC2 function.

In search for constructs suitable for structural studies, we overexpressed a truncated minimally active EZH1-containing PRC2, EZH1–EED–SUZ12(VEFS), in *Saccharomyces cerevisiae* for crystallization. Unexpectedly, we found the majority of SUZ12 from the purified complex is phosphorylated at residue S583 according to the mass spectrometry result (Supplementary Fig. 5), likely by endogenous yeast CK2. In addition, human CK2 was able to specifically phosphorylate S583 within the truncated PRC2-EZH1 minimal complex pretreated by λ protein phosphatase, confirming the CK2 kinase specificity in this context (Supplementary Fig. 6).

We determined the 3.0 Å crystal structure of this minimal complex, which successfully captures residue S583 in the phosphorylated state (Fig. 3b, Supplementary Fig. 4c and Supplementary Table 1). Upon phosphorylation, the flexible loop harboring S583 and neighboring acidic residues dramatically change conformation, becoming engaged with the SET domain of EZH1 (Fig. 3b and Supplementary Movie 1). In parallel, phosphoserine induces self-packing of the N-terminal portion of the SUZ12(VEFS), which contacts EED and the SET domain simultaneously (Fig. 3b and Supplementary Movie 1). EZH1 and EZH2 share a nearly identical SET domain (Supplementary Fig. 7), and therefore structural analysis on the EZH1-containing PRC2 here likely applies to the equivalent EZH2-containing complex.

The core of the SUZ12 loop undergoing phosphorylation-induced conformational change is a motif of three acidic residues, D582-S583p-E584, which makes extensive interactions with one lysine residue, K684, protruding from the SET domain: both the phosphate group of S583p and the carboxyl group of D582 side chain form hydrogen bonds with the amino group of K684 side chain, whereas the carboxyl group of E584 side chain mediates an additional hydrogen bonding interaction with the main chain amine of K684 (Fig. 3c, d). Residues H567 and S568 of the VEFS domain of SUZ12 also contact the phosphate group (Fig. 3c, d). Other residues helping shape the local conformation include K612 and L615 of the SET domain and E586, D588, and R593 of the VEFS domain (Fig. 3d). Notably, residue K684 of the SET domain belongs to a single turn helix partially lining the SAM-binding pocket at the enzyme active site (Fig. 3c). We predicted that the interaction network around K684 organized by the phosphoserine may enhance PRC2 enzymatic activity by stabilizing the SET domain and facilitating SAM binding. Accordingly, the S583-containing regulatory loop of SUZ12 is hereinafter referred to as the phosphorylation-dependent stimulation (PDS) loop (Figs. 1a and 3b).

## S583 phosphorylation enhances enzymatic activity and nucleosome binding of PRC2 in vitro

Mutations of EZH2 residue K683 (the equivalent of EZH1 residue K684) and SUZ12 residues H567, S568, D582, and S583 were all found in cancer cells, including established cancer cell lines and patient samples (Supplementary Fig. 8)[57] (cancer.sanger.ac.uk/cosmic), suggesting the molecular interactions mediated by these residues may help maintain normal PRC2 function (Fig. 3c, d). In consistence, minimal complexes containing a K684A single mutation on EZH1 or an H567A/S568A double mutation on SUZ12 displayed exceedingly reduced methyltransferase activities towards mononucleosome substrates (Fig. 4a), likely due to disruption of the phosphoserine-centered interactions. Similar results were obtained for the same set of mutations in the context of the EZH2-containing minimal PRC2 complex (Fig. 4b).

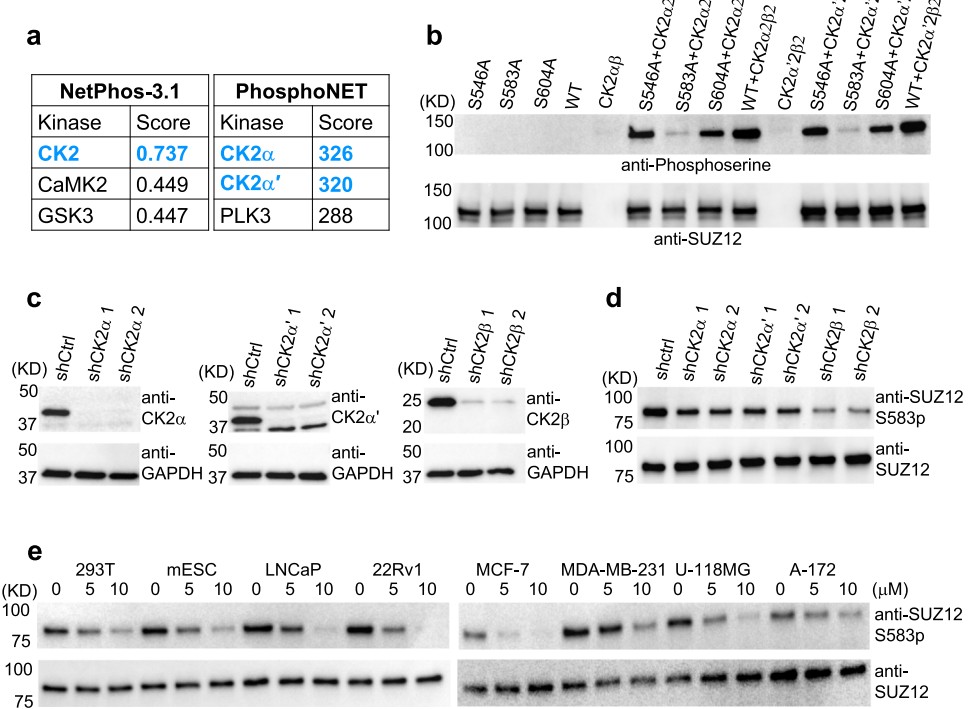

**Fig. 2 | CK2 is the kinase for the phosphorylation of S583 of SUZ12. a** Kinase prediction by web servers NetPhos 3.1 and PhosphoNET. The peptide motif around S583 was used for the prediction. The top three hits are listed in each case with CK2 highlighted in blue. **b** In vitro phosphorylation assay. CK2 complexes were expressed in HEK293T cells and GST-tagged full-length SUZ12 WT and mutants were expressed in bacteria. Total serine phosphorylation was measured by an anti-phosphoserine antibody (uncropped gel images of this figure are shown in Supplementary Figs. 16 and 17). A representative of three independent experiments is shown. **c** Stable knockdown of CK2 subunits a, a', and b in NT2/D1 cells. Two independent shRNAs were tested for knockdown efficiency. A representative of two independent experiments is shown. **d** S583 phosphorylation in NT2/D1 in the presence of CK2 knockdown. A representative of three independent experiments is shown. In **d** and **e**, immunoprecipitates of the anti-SUZ12 antibody were used for the detection of S583 phosphorylation. **e** Effect of chemical inhibition of CK2 kinase activity on S583 phosphorylation. Cell lines were treated with indicated concentrations of CX4945 for 24 h. Source data are provided as a Source Data file.

To examine the contribution of the interacting residues to enzymatic activity in a more complete system, we purified ectopically expressed EZH2-containing wild-type (WT) and mutant PRC2-5m complexes from HEK293T cells (Supplementary Fig. 9). No endogenous phosphorylated SUZ12 was detected in the purified SUZ12S583A mutant complex (Supplementary Fig. 10). When SUZ12 harbors the S583A single mutation and thus lacks phosphorylation at this site, histone methylation was severely compromised (Fig. 4c). All methylation states were affected by the S583A mutation (Supplementary Fig. 11). In comparison, the S583D phosphomimetic mutant complex did not display a defect in catalysis (Fig. 4c). In addition, the K683A mutation of EZH2 and the H567A/S568A mutation of SUZ12 also noticeably impaired the enzymatic activity in this context (Fig. 4c). More directly, CK2-mediated in vitro re-phosphorylation of λ phosphatase-treated WT PRC2-5m pronouncedly enhanced histone methylation (Fig. 4d).

To dissect how S583 phosphorylation facilitates catalysis, we performed a steady-state enzymology study with PRC2-5m containing WT or S583A mutant SUZ12 using histone peptide substrates (Fig. 4e). Assays were conducted under both histone peptide-saturating and SAM-saturating conditions (Fig. 4e). As indicated by the $K_m$ values changing from 0.5 to 2.9 µM, loss of S583 phosphorylation most profoundly affected SAM binding to PRC2, whereas histone peptide binding was only moderately weakened (Fig. 4e). This is in line with the structural observation that S583 phosphorylation stabilizes the SAM-binding pocket (Fig. 3c). In comparison, enzyme turnover $k_{cat}$ did not seem to be affected by the mutation (Fig. 4e).

To check if phosphorylation of S583 of SUZ12 plays a role in PRC2 binding to nucleosomes, we assembled mononucleosomes with a

biotinylated DNA and performed avidin bead pulldown assays. Compared to the WT counterpart, PRC2-5m containing the S583A mutation displayed markedly reduced interaction with nucleosomes; however, in the absence of histone H3 tail (residues 1–27), nucleosome binding was equally diminished for the WT and mutant PRC2-5m (Fig. 4f), suggesting S583 phosphorylation may be necessary for optimal binding of enzyme active site to the histone tail in the nucleosomal context, especially when SAM concentration is not saturating but likely limiting. Congruently, nucleosomes were bound less tightly by λ phosphatase-treated WT PRC2-5m, compared to the same complex re-phosphorylated by CK2 in vitro (Fig. 4g). To gain a quantitative view of nucleosome binding, we performed native gel shift assays. The nucleosome binding affinity of the S583A mutant PRC2-5m complex was reduced by roughly two folds compared to that of the WT complex (Fig. 4h and Supplementary Fig. 12a, b). Correspondingly, nucleosome binding by PRC2-5m was also impaired in the absence of the N-terminal tail of histone H3 (Supplementary Fig. 12c).

## S583 phosphorylation promotes reporter gene repression

A transient expression luciferase gene reporter system was previously established to recapitulate PRC2-dependent gene repression in cells[58]. Here, we used a similar system to examine the role of S583 phosphorylation in reporter gene repression in an engineered HEK293T cell line with endogenous SUZ12 knocked out (HEK293T$^{\Delta SUZ12}$)[11]. Specifically, a "6×GAL4UAS" cassette was inserted upstream of the thymidine kinase (TK) promoter that controls the luciferase reporter gene. SUZ12 protein fused to the GAL4 DNA binding domain (GAL4DBD) was transiently expressed in HEK293T$^{\Delta SUZ12}$ cells together with the reporter plasmid. GAL4DBD

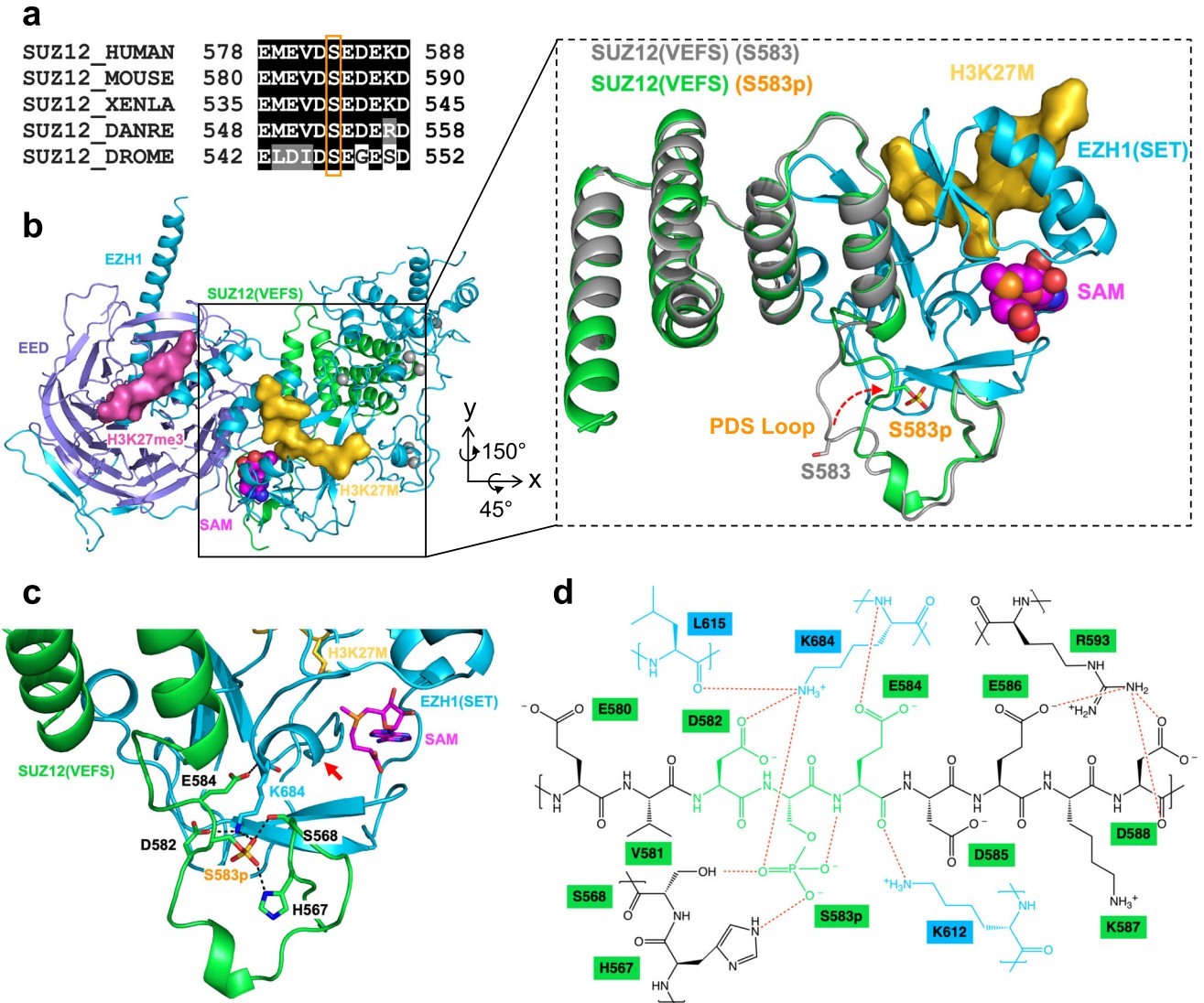

**Fig. 3 | S583 phosphorylation stabilizes PRC2 active site. a** Alignment of SUZ12 sequences around residue S583 in several model organisms. **b** Structure of the minimal PRC2-EZH1 complex with a phosphorylated S583. The overall structure is provided on the left with a close-up view on the right. Protein subunits, peptides, and the cofactor included in the crystal structure are color-coded and labeled. A previously reported structure of the minimal PRC2-EZH2 complex that lacks S583 phosphorylation (PDB 5HYN) is superimposed on the current structure in the close-up view and is colored in gray. Conformational change of the PDS loop induced by S583 phosphorylation is indicated by the red arrow. **c** Phosphoserine-centered interaction network. Interacting residues are shown as sticks. The red arrow indicates the single turn helix of the SAM-binding pocket. Some interacting residues are omitted for clarity. **d** 2D schematic of the interaction network. Interacting residues from the SET domain are colored in blue, the DSpE core motif from the PDS loop is colored in green, and the rest are colored in black.

recruits ectopically expressed SUZ12 in complex with other endogenous PRC2 subunits to the TK promoter (Fig. 5a).

We first tested the dependence of reporter gene repression on PRC2 enzymatic activity. Compared to the GAL4DBD alone control construct, full-length SUZ12 was sufficient to confer reporter gene repression (Fig. 5b and Supplementary Fig. 13). The VEFS domain of SUZ12 essential for the assembly of the minimally active PRC2, EZH2–EED–SUZ12(VEFS), mediated comparable gene repression (Fig. 5b and Supplementary Fig. 13), suggesting that accessory subunits of PRC2 are largely dispensable for this artificial targeting system and thus will not complicate data interpretation. A highly specific PRC2 enzyme inhibitor EPZ6438 relieved reporter gene repression in a dose-dependent manner in both contexts (Fig. 5b and Supplementary Fig. 13), indicating the observed reporter gene repression was correlated with PRC2 enzymatic activity in cells.

A W555C mutation within the VEFS domain of *Drosophila* SU(Z)12 was previously shown to cause a dramatic decrease in PRC2 enzymatic activity in vitro[55]. In the current assay, the equivalent W591C mutation of human SUZ12 led to reporter gene derepression (Fig. 5c and Supplementary Fig. 13). Similar to this positive control, the S583A mutation of SUZ12 also derepressed the reporter gene when present in either the full-length or minimal construct (Fig. 5c and Supplementary Fig. 13), which suggests S583 phosphorylation can directly promote reporter gene repression in cells, likely by enhancing PRC2 enzymatic activity. In support of the role of S583 phosphorylation, the S583D phospho-mimetic mutation was not found to compromise the reporter gene repression (Supplementary Fig. 14).

**Loss of S583 phosphorylation disturbs PRC2 targeting and H3K27me3 deposition in mESCs and impairs cell identity maintenance during mESC differentiation**

PRC2 is known to be required for proper differentiation of mESCs, but dispensable for self-renewal and pluripotency of these cells[59,60]. In mESCs, mSUZ12 is substantially phosphorylated at residue S585, the

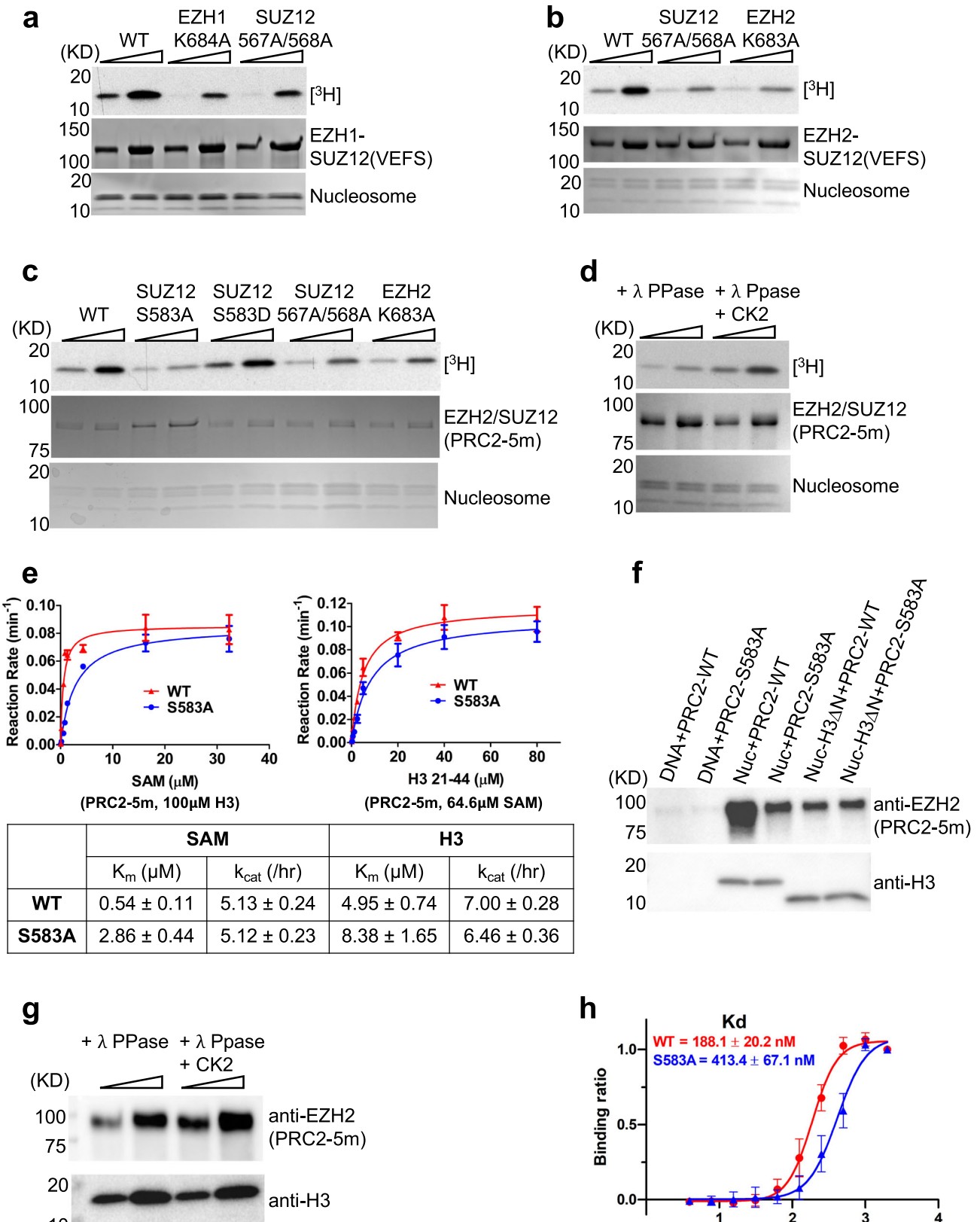

equivalent of residue S583 of human SUZ12 (Fig. 1b). To study how S583 phosphorylation impacts PRC2 function in vivo, we re-expressed 3×FLAG-tagged human SUZ12 WT (SUZ12^WT) and S583A (SUZ12^S583A) mutant that eliminates phosphorylation in a mSUZ12 knockout (KO) mESC line[61], using lentiviral vectors (Fig. 6a). Pluripotent mESCs were maintained in serum-free 2i media[62,63]. An equal amount of WT and

mutant SUZ12 was bound to EZH2 in an anti-EZH2 co-immunoprecipitation (Co-IP) assay (Fig. 6b), indicating the phosphoserine-centered interactions between SUZ12 and EZH2 are not essential for PRC2 assembly. In addition, PRC2 containing SUZ12^S583A displayed a slightly weaker association with bulk chromatin in mESCs than the WT PRC2 (Fig. 6c), suggesting a possible chromatin binding defect.

**Fig. 4 | S583 phosphorylation promotes PRC2 function in vitro. a** Radioactive methyltransferase assay with the PRC2-EZH1 ternary complex (EZH1–EED–SUZ12 (VEFS)) and mononucleosome substrates. Assays were performed using 150 and 450 nM of the WT and mutant enzymes (uncropped gel images of this figure are shown in Supplementary Figs. 16 and 17). A representative of three independent experiments is shown. **b** The same as **a**, except that the PRC2-EZH2 ternary complex (EZH2–EED–SUZ12(VEFS)) was used. A representative of three independent experiments is shown. **c** Radioactive methyltransferase assay with mononucleosome substrates and PRC2-5m WT and mutant holo complexes expressed in HEK293T cells. 50 and 100 nM of WT and mutant enzymes were used. A representative of three independent experiments is shown. **d** Radioactive methyltransferase assay with λ phosphatase and CK2-treated PRC2-5m. WT PRC2-5m used in this assay was expressed in Sf9 cells. λ phosphatase-treated PRC2-5m was subjected to size exclusion chromatography to remove λ phosphatase. Dephosphorylated PRC2-5m was re-phosphorylated by human CK2 in vitro. 50 and 100 nM of the dephosphorylated and re-phosphorylated PRC2-5m were used for the methyltransferase assay and compared. A representative of two independent

experiments is shown. **e** Steady-state enzymology study of PRC2-5m WT and S583A mutant. Assays performed under the substrate peptide-saturating condition are shown on the left and assays under the SAM-saturating condition are on the right. GraphPad Prism was used to fit the data and derive $K_m$ and $k_{cat}$ values. $n = 3$ independent enzymatic reactions. Error bars represent mean ± SEM. **f** Nucleosome binding assay. Biotinylated nucleosomal DNA was generated by PCR with a biotin-labeled primer. Bound WT and mutant PRC2-5m expressed in HEK293T cells are indicated by anti-EZH2 signals. Anti-H3 signals for H3 and H3ΔN are controls for the bait. H3ΔN lacks residues 1–27 of histone H3. A representative of two independent experiments is shown. **g** The same as **f**, except that dephosphorylated and re-phosphorylated Sf9-expressed PRC2-5m were used for the binding assay. Two amounts of the bound PRC2-5m (1× and 3×) were loaded on the gel. A representative of two independent experiments is shown. **h** Native gel shift nucleosome binding assay. Mononucleosomes and HEK293T-expressed PRC2-5m WT and mutant were used for the binding assay. $K_d$ values were calculated based on $n = 3$ independent gel shift assays. Error bars represent mean ± SEM. Source data are provided as a Source Data file.

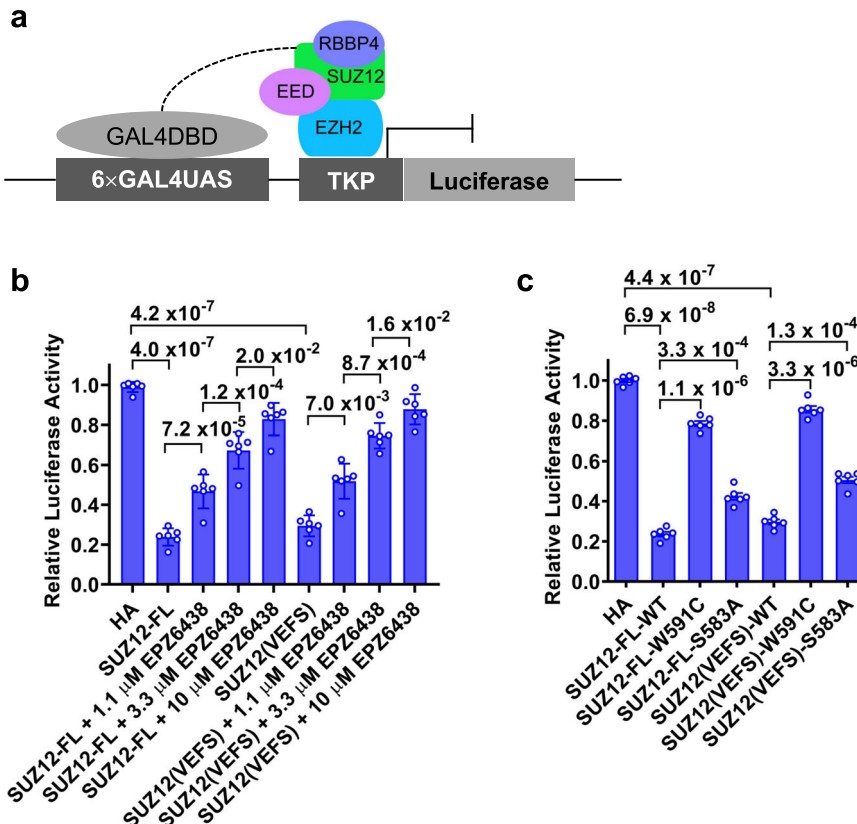

**Fig. 5 | S583 phosphorylation facilitates reporter gene repression. a** Schematic of GAL4-based reporter gene repression assay. **b** PRC2 enzymatic activity-dependent reporter gene repression. GAL4DBD-HA-tagged SUZ12-FL or SUZ12(-VEFS) was transiently expressed in HEK293T cells that lack endogenous SUZ12. EPZ6438 is a selective enzyme inhibitor of PRC2. In **b** and **c**, assays were performed on three different days with the measurement of two replicate wells recorded each time. Signals were normalized to. GAL4DBD-HA negative control. *p* values were

derived from two-sided *t*-tests performed in Microsoft Excel. $n = 6$ biologically independent experiments. Error bars represent mean ± SEM. **c** Effect of S583A loss-of-phosphorylation mutation on reporter gene repression. Assays were performed in the context of SUZ12-FL or SUZ12(VEFS). W591C is a known mutation within the SUZ12(VEFS) that disrupts PRC2 enzymatic activity. Source data are provided as a Source Data file.

To assess PRC2 recruitment and H3K27me3 deposition on individual gene loci in mESCs expressing SUZ12[WT] or SUZ12[S583A], we carried out chromatin immunoprecipitation (ChIP)-qPCR experiments focusing on known PRC2 targets. The active pluripotent gene *NANOG* served as a non-target negative control. As shown by the anti-FLAG ChIP data, the chromatin recruitment of SUZ12 was impaired by the S583A mutation on members of *HOX* gene clusters, *HOXA7* and *HOXD12*, where PRC2 is highly enriched (Fig. 6d). Similar reduction in chromatin binding was also observed for the mutant on other lineage marker

genes with varying degrees of PRC2 enrichment, including *GATA4*, *FGF5*, and *NESTIN* (Fig. 6d). In line with the defect in chromatin binding, H3K27me3 levels were also affected by the mutation on many of these gene loci (Fig. 6e).

We next investigated the ability of PRC2 to maintain the differentiated state of mESCs, using a recently reported replating assay[64]. mESCs expressing SUZ12[WT] or SUZ12[S583A] were differentiated to form embryoid bodies (EBs), which were subsequently dissociated into single cells; these single cells were next replated in 2i

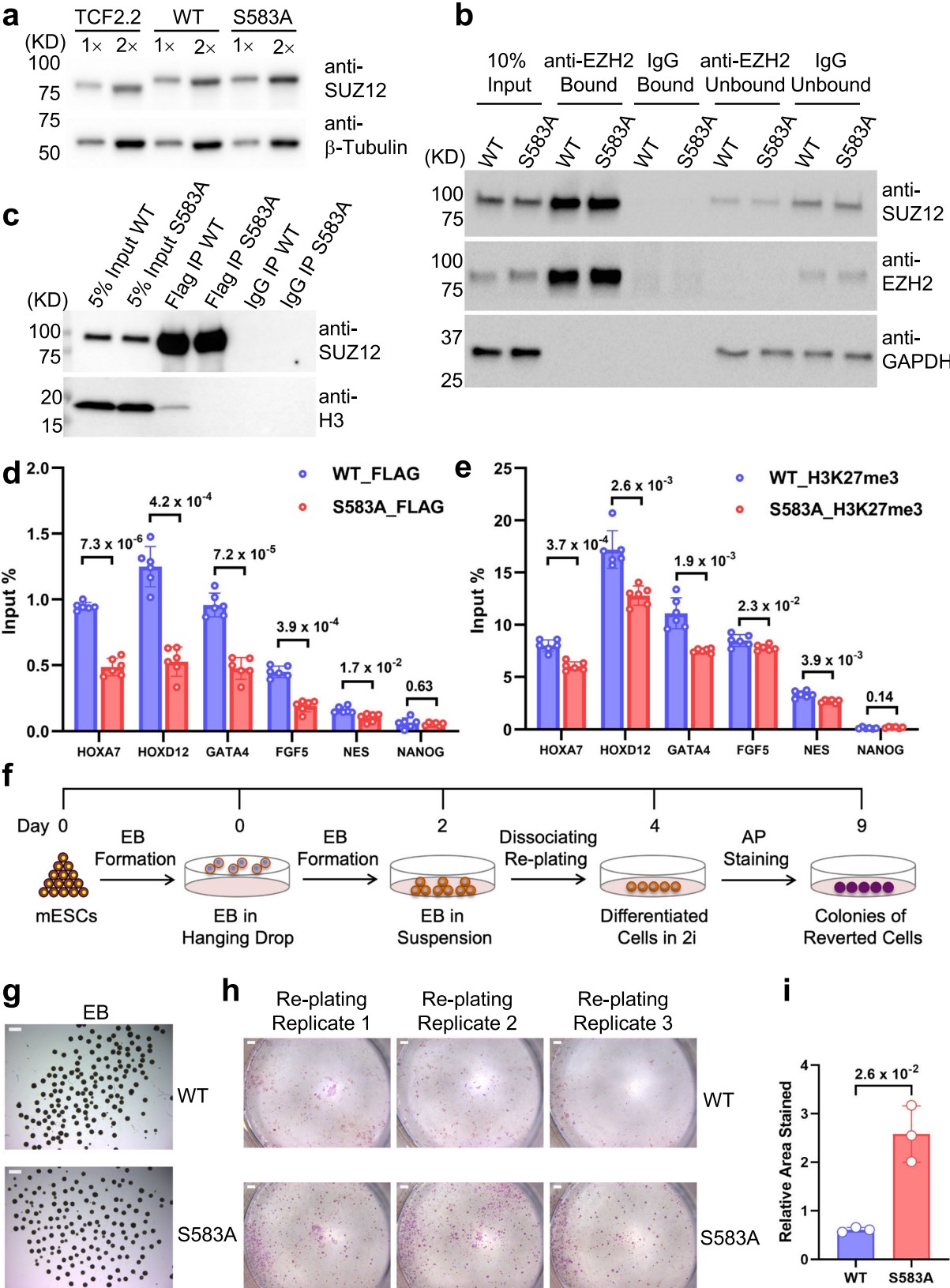

media, a growth condition that challenges the maintenance of the differentiated cell identity (Fig. 6f). EBs formed by the WT and mutant mESCs were indistinguishable in morphology (Fig. 6g), indicating mESCs lacking S583 phosphorylation retains the capacity to differentiate, despite the apparent defect in PRC2 targeting and H3K27me3 deposition in the pluripotent state of mESCs (Fig. 6d, e).

A drastic phenotype appeared when differentiated cells from these EBs were replated in 2i media: a large number of SUZ12^S583A-containing cells were reverted to a pluripotent stem cell state as shown by alkaline phosphatase (AP) staining, whereas cell identity reversion was only sporadic for SUZ12^WT-containing cells (Fig. 6h, i and Supplementary Fig. 15), suggesting the phosphorylation of S583 of

**Fig. 6 | S583 phosphorylation is important for PRC2 recruitment, H3K27me3 deposition, and cell identity maintenance. a** SUZ12 expression levels in the parental and engineered mESCs. SUZ12 from the parental mESC line and engineered mESC lines with the re-expression of 3×FLAG-SUZ12-FL-WT or 3×FLAG-SUZ12-FL-S583A was checked by western blot (uncropped gel images of this figure are shown in Supplementary Fig. 18). A representative of two independent experiments is shown. **b** Integrity of PRC2 assembly. Anti-EZH2 antibody was used to capture re-expressed WT and mutant SUZ12 by co-immunoprecipitation. Both bound and unbound fractions were analyzed by western blot for SUZ12 (prey), EZH2 (bait), and GAPDH (loading control). Rabbit IgG was a negative control. A representative of two independent experiments is shown. **c** PRC2 binding to bulk chromatin. FLAG immunoprecipitation was used to capture FLAG-tagged SUZ12 and associated chromatin fragments generated by sonication. Bound chromatin is indicated by anti-H3 signals. A representative of two independent experiments is shown. **d** Anti-FLAG ChIP-qPCR. Binding of WT and S583A mutant SUZ12 to known PRC2 targets was compared. In **d** and **e**, two independent ChIP experiments were performed each with three qPCR replicates. *p* values were derived from two-sided *t*-tests performed in Microsoft Excel. *NANOG* is a negative control. *n* = 6 independent experiments. Error bars represent mean ± SEM. **e** Anti-H3K27me3 ChIP-qPCR. H3K27me3 deposition at known PRC2 targets in mESCs expressing WT or S583A mutant SUZ12 was compared. **f** Schematic of the replating assay. **g** EB formation. Morphology of EBs differentiated from mESCs expressing WT or S583A mutant SUZ12 was compared. Scale bar stands for 1 mm. A representative of three independent experiments is shown. **h** Reversion of the differentiated cell identity. Cells dissociated from EBs expressing WT or S583A mutant SUZ12 were replated in 2i media and checked for pluripotency by AP staining. Replating assays were performed three times using cells from three independent EB formation experiments. Scale bar stands for 1 mm. **i** Quantification of AP staining. Relative areas stained by AP were quantified in ImageJ. *p* values were derived from two-sided *t*-tests performed in Microsoft Excel. *n* = 3 biologically independent experiments. Error bars represent mean ± SEM. Source data are provided as a Source Data file.

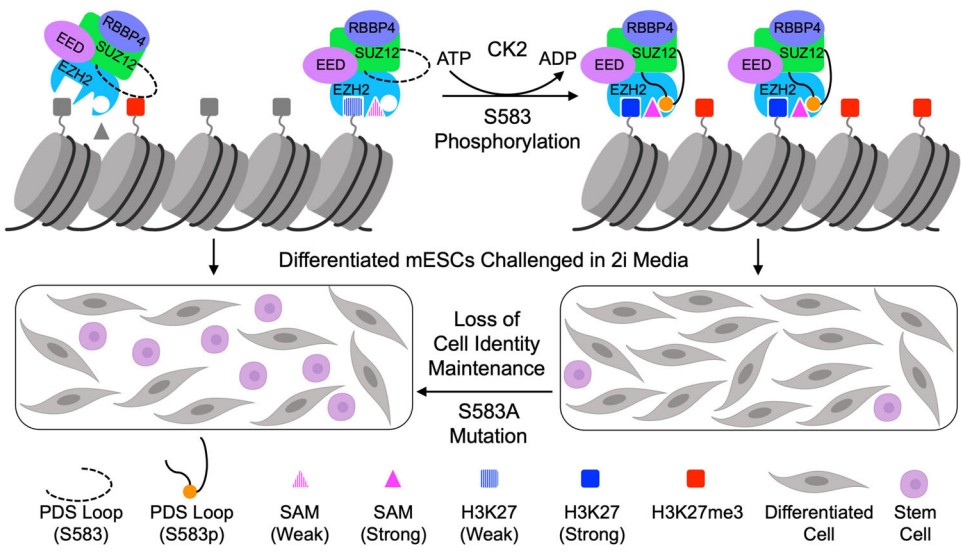

**Fig. 7 | A model of PRC2 function promoted by the phosphorylation of S583 of SUZ12.** Cartoons illustrate how SUZ12 phosphorylation stabilizes enzyme active site and promotes PRC2 function. S583 phosphorylation induces conformational change of the PDS loop of SUZ12, stabilizes the SAM-binding pocket, and converts a weak binding state of SAM to a strong binding state. This also facilitates histone substrate H3K27 binding. PRC2 recruitment and H3K27me3 deposition are enhanced in this way. Cell identity maintenance is compromised when differentiated mESCs are challenged in 2i media in the absence of S583 phosphorylation. Created with BioRender.com.

SUZ12 is essential for PRC2 function in maintaining cell identity during mESC differentiation.

## Discussion

PRC2 sets an epigenetic threshold for maintaining cell identity[2]. In supporting this pivotal function, PRC2 enzymatic activity is subjected to complex cellular regulation. In the current work, we reveal a unique phosphorylation-dependent mechanism that stimulates PRC2 enzymatic activity. Our structural study provides direct evidence for how a posttranslational modification of a PRC2 core subunit may regulate enzyme function. Upon phosphorylation of residue S583 in the SUZ12(VEFS) domain, the PDS loop undergoes a dramatic conformational change: it transitions from a partially disordered state to become engaged with the catalytic SET domain, stabilizing the enzyme active site (Fig. 7). The PDS loop is an addition to a collection of flexible structural elements dictating distinct functional states of PRC2. Other notable examples include the stimulation-responsive motif (SRM) of EZH2 that bridges the stimulating signal from H3K27me3 to the SET domain[8,9], the bridge helix of EZH2 that connects nucleosomal substrates and the SET domain[48], and the C2 domain of SUZ12 that associates with the accessory subunits MTF2 and PHF19 of PRC2.1 and AEBP2 of PRC2.2 and directly mediates PRC2 dimerization crucial for chromatin binding[10,11].

In analyzing the structural plasticity and phosphorylation-dependent interactions of the PDS loop, we noticed that residue K684 of the SET domain of EZH1 (the equivalent of residue K683 of EZH2) close to the SAM-binding pocket is stabilized by an acidic motif of SUZ12 centering on the phosphoserine (Fig. 7). Consistently, our enzymology data using the WT and S583A mutant PRC2-5m confirmed that SAM binding was severely compromised for the mutant (Fig. 4e). In addition, when SAM concentration is limiting, PRC2 binding to nucleosomal substrates is also impaired in the absence of S583 phosphorylation (Fig. 4f–h), likely due to the structural coupling of SAM and histone H3 tail binding to the enzyme active site. Accordingly, diminished PRC2 enzymatic activity caused by disruption of the phosphoserine-centered interactions can be readily rationalized by weakened binding of PRC2 to SAM and histone tail (Fig. 4a–d). Intracellular availability of SAM as a critical metabolite is known to influence histone methylation and gene regulation[65]. In this regard, phosphorylation of S583 of SUZ12 may serve as a cellular mechanism to maintain chromatin occupancy and enzymatic activity of PRC2 in case of metabolic perturbations.

mESC differentiation provides a valuable system for studying PRC2 function in vivo. Self-renewal and pluripotency of mESCs are not changed even by some extreme alterations of PRC2, including partial or full deletion of SUZ12, which results in redistribution or complete loss of H3K27me3, respectively[61]. We found that the majority of SUZ12 in mESCs are phosphorylated at residue S583 and that the S583A mutation is sufficient to reduce PRC2 enrichment on target genes, which is also accompanied by a decrease in H3K27me3 deposition (Figs. 1b and 6d, e). A prominent cell identity reversion phenotype arises when differentiated mESCs dissociated from EBs are replated in 2i media promoting pluripotency[64]. The number of SUZ12^S583A-expressing mESCs reverting to the pluripotent state greatly exceeds that of SUZ12^WT-expressing mESCs (Fig. 6h, i), suggesting that PRC2 function in cell identity maintenance is compromised by the lack of S583 phosphorylation (Fig. 7). A hypomorphic mutation of EZH2 also impedes cell identity maintenance during mESC differentiation, and it is proposed that full methylation of H3K27 is required for stable commitment to differentiation[64]. In this regard, S583 phosphorylation can be a missing piece of the puzzle of cell identity maintenance by PRC2. It is not impossible that defects in cell differentiation not revealed by the visual inspection of EBs from the SUZ12^S583A-expressing mESCs may also exist. In addition, it remains to be studied if the level of S583 phosphorylation changes during early differentiation or in other developmental stages, although it does appear to vary in some cancer cell lines (Fig. 1e and Supplementary Fig. 3).

SUZ12 was previously found in the CK2 interactome in mitotic HEK293T cells[66]. In this study, we showed that CK2 is the kinase responsible for phosphorylation of S583 of SUZ12 (Fig. 7). This finding connects a widespread cell signaling event known to regulate cell proliferation and apoptosis to a key epigenetic mechanism preserving cell identity. Our data also predict that clinically relevant CK2 inhibitors may impair PRC2 function indirectly by inhibiting CK2-mediated S583 phosphorylation. CK2 is a ubiquitous and constitutively active kinase, and CK2 expression is often elevated in cancer cells[39]. This raises the question of whether and how S583 phosphorylation is regulated under physiological conditions. In addition, given that CK2 serves as a subunit of PRC1.3 and PRC1.5 and that PRC1 and PRC2 co-occupy target loci in Polycomb chromatin domains, it would be interesting to explore if S583 of SUZ12 is phosphorylated in the context of these variant PRC1 complexes, which would add another mechanistic link between the two major complexes of the Polycomb repressive system.

# Methods

## Cell culture

HEK293T, A172, MDA-MB-231, and U118MG cell lines were cultured in DMEM (Sigma, Cat No. D5796) supplemented with 10% FBS (Sigma, Cat No. 2442) and 1× penicillin-streptomycin (Sigma, Cat No. P0781). LNCaP and 22RV1 cells were cultured in RPMI 1640 (ATCC, Cat No. 30−2001) supplemented with 10% FBS and 1× penicillin-streptomycin. NT2/D1 cells were cultured in DMEM (ATCC, Cat No. 30−2002) supplemented with 10% FBS and 1× penicillin-streptomycin. MCF-7 cells were cultured in EMEM (ATCC, Cat No. 30−2003) supplemented with 10 μg/ml human insulin (Sigma, Cat No. 91077C), 10% FBS, and 1× penicillin-streptomycin. MCF10A cells were cultured in the Mammary Epithelial Cell Growth Medium (Sigma, Cat No. C-21010) supplemented with 1× penicillin-streptomycin. BT-474 cells were cultured in RPMI 1640 supplemented with 20% FBS, 10 μg/ml human insulin, 2 mM L-glutamine, and 1× penicillin-streptomycin. mESCs were cultured in 2i media, containing a 1:1 mix of DMEM/F12 (GIBCO, Cat No. 11320033) and Neurobasal media (GIBCO, Cat No. 21103049), 1× penicillin-streptomycin (Sigma, Cat No. P0781), 0.05% BSA (Fisher, Cat No. 15260037), 100 μM BME (Sigma, Cat No. M3148), 0.5× GlutaMax (GIBCO, No. 35050061), 0.5% N-2 supplement (GIBCO, Cat No. 17502048), 1% B-27 Supplement (GIBCO, Cat No. 17504044), 3 μM GSK

inhibitor CHIR99021 (Cayman Chemical, Cat No. 131225), 1 μM MEK inhibitor PD0325901 (Cayman Chemical, Cat No. 130345), and LIF produced in the lab. The activity of the homemade LIF was assayed based on marker gene expression and morphology of mESC colonies.

## Antibodies

The following commercial antibodies were used in this study: rabbit anti-SUZ12 (Cell Signaling, Cat No. 3737, 1:1000 dilution for western blot), rabbit anti-CK2α (GeneTex, Cat No. GTX107897, 1:500 dilution for western blot), rabbit anti-CK2α′ (Bethyl, Cat No. A300-199A, 1:500 dilution for western blot), rabbit anti-CK2β (Bethyl, Cat No. A301−984A, 1:500 dilution for western blot), rabbit anti-phosphoserine (Abcam, Cat. No. ab9332, 1:500 dilution for western blot), mouse anti-GAPDH (Invitrogen, Cat No. MA515738, 1:1000 dilution for western blot), rabbit anti-EZH2 (Cell Signaling, Cat No. 5246, 1:1000 dilution for western blot), rabbit anti-H3 (Cell Signaling, Cat No. 4499, 1:5000 dilution for western blot), rabbit anti-HA tag (Cell Signaling, Cat No. 3724, 1:1000 dilution for western blot), mouse anti-FLAG tag (Sigma, Cat No. F1804, 1:1000 dilution for western blot and 1:500 dilution for ChIP), rabbit anti-H3K27me3 (Cell signaling, Cat No. 9733, 1:1000 dilution for western blot and 1:200 dilution for ChIP), rabbit anti-H3K27me2 (Millipore, Cat No. 07−452, 1:500 dilution for western blot), rabbit anti-H3K27me1 (Millipore, Cat No. 07−448, 1:500 dilution for western blot), and rabbit anti-β-Tubulin (Cell Signaling, Cat No. 2128, 1:1000 dilution for western blot). Rabbit antibody specific for SUZ12S583p was generated by the Animal Resource Center (ARC) of UT Southwestern Medical Center using the KLH conjugated peptide: KLH-CQEMEVD-[phospho-S]-EDEKDPE. The anti-SUZ12S583p antibody in rabbit sera was purified by peptide affinity columns containing crosslinked apo or phosphoserine peptides. The anti-SUZ12S583p antibody was diluted by 1000 folds for western blot.

## Re-expression of SUZ12 in SUZ12 knockout mESCs with lentiviral vectors

SUZ12 knockout mESC line is a generous gift from Dr Kristian Helin (Institute of Cancer Research)[61]. SUZ12 was re-expressed in the knockout cell line using lentiviral vectors. cDNA sequence encoding human WT or S583A mutant SUZ12 with an N-terminal 3×FLAG tag was subcloned into the pCDH-EF1α-MCS-IRES-Puro vector using XbaI and EcoRI restriction sites. For lentivirus production, the pCDH-EF1α−3×FLAG-SUZ12 WT or S583A plasmid (5 μg), psPAX2 (5 μg), and pVSV-G (0.5 μg) were co-transfected into HEK293T cells at ~70% confluence. The medium containing lentivirus particles was harvested 48 h post transfection and centrifuged at 200 g for 10 min. The supernatant was passed through a 0.45-μm filter and precipitated by 1/3 volume of Lenti-X concentrator (Takara, Cat No. 631231), followed by mixing on a nutator for 30 min at 4 °C and then centrifugation at 1500 g for 45 min. The Lentivirus particles were resuspended in the 2i condition medium, aliquoted, flash-frozen by liquid nitrogen, and stored at −80 °C till transduction.

For transduction, 1 × 10^5 SUZ12 knockout mESCs were seeded at a 6-well plate, 24 h before transduction. Cells were transduced by lentiviruses expressing respective SUZ12 constructs together with 10 μg/ml polybrene (Sigma, Cat No. TR-1003-G). 1 μg/ml puromycin (Sigma, Cat No. P8833) was supplemented to the growth media 48 h post transduction. After 72 h, mESCs were diluted and seeded into 96-well plates in the presence of 1 μg/ml puromycin. Single-cell colonies expressing comparable amounts of WT and S583A mutant SUZ12 were identified by western blotting and were frozen for downstream analysis.

## Stable knockdown of CK2

Lentiviruses for stable knockdown of CK2 components, CK2α, CK2α′, and CK2β, were generated using the pLKO.1 lentiviral vector that expresses corresponding shRNAs (Sigma). shRNA sequences are provided in Supplementary Table 2. The same lentivirus production

protocol described above was followed. NT2/D1 cells were seeded onto 6-well plates at 30% confluence. Lentiviruses were added into the cell culture together with 10 µg/ml polybrene 24 h post cell seeding. After 48 h of transduction, cells were selected in the growth medium containing 1 µg/ml puromycin for 6 days with a medium change every 48 h. Cells resistant to puromycin were lysed for western blot to detect the knockdown efficiency and stored for downstream analysis.

## Recombinant protein expression and purification

The ternary human PRC2-EZH1 and PRC2-EZH2 complexes (EZH1/2–EED–SUZ12(VEFS)) used for enzymatic assays contained a full-length EZH1/2 (residues 1–747 and 1–746) fused to the VEFS domain of SUZ12 (residues 543–695) and a full-length EED (residues 1–441). cDNA corresponding to the His$_6$–2×Protein A-TEV-EZH1/2-LVPRGS-SUZ12(VEFS) fusion construct was subcloned into the p416GAL1 vector with a URA marker. EED was subcloned into the p415-GAL1 vector (LEU marker). The minimal complex used for crystallization contained the following modifications: residues 188–229 of EZH1 were replaced by a GGGSGGGSGGGS linker sequence, residues 353–413 of EZH1 were deleted, residues 492–496 of EZH1 were replaced by a GGSGG linker sequence, and residues 1–77 of EED was replaced by a StrepII tag. The two plasmids were co-transformed into an *S. cerevisiae* CB010 strain, followed by selection on a synthetic drop-out medium plate lacking uracil and leucine. Starters of transformed yeast cells were grown in synthetic drop-out media with 2% raffinose. Protein expression was induced by 2% galactose in YP media for about 20 h. The minimal complex was purified by IgG-sepharose and eluted from the resin by TEV protease cleavage. The protein complex was further purified by size exclusion chromatography on Superdex 200. Protein complex purity was assessed by SDS–PAGE.

WT and mutant human PRC2-5m complex (EZH2–EED–SUZ12–RBBP4–AEBP2) was expressed in HEK293T cells. Briefly, cDNAs of HA-EZH2, His$_6$-EED, and HA-RBBP4 were inserted into the pCS2+ vector. cDNA corresponding to 2×Protein A-TEV-SUZ12-HA was inserted into the pCS2+ vector. cDNA corresponding to 2×Protein A-3C-AEBP2 (residues 1–295) was inserted into the pCS2+ vector. These five plasmids were co-transfected into HEK293T cells at ~70% confluence by polyethylenimine (PEI). Cells were harvested 48 h post transfection. Protein complexes were purified by IgG affinity resin and released by TEV and HRV-3C protease cleavage overnight at 4 °C. Protein complex purity was confirmed by SDS–PAGE. WT human PRC2-5m complex expressed in Sf9 cells was purified as described previously[11].

CK2α and CK2α' cDNAs were tagged at the 5' ends with a 2×Protein A tag followed by a TEV protease site and were inserted into the pHEK293 ultra expression vector (Takara). CK2β was tagged with a SUMO tag at the 5'-end and was cloned into the pHEK293 ultra vector as well. HEK293T cells were co-transfected by PEI with the plasmids expressing CK2α plus CK2β or CK2α' plus CK2β. Cells were harvested 48 h post transfection. CK2 complexes were purified by IgG affinity column, and protein purity was assessed by SDS–PAGE.

To prepare GST-SUZ12 proteins from bacterial expression, the cDNA sequence encoding full-length human SUZ12 (1–739) was subcloned into the pGEX-4T-1 vector. Alanine mutations were introduced by site-directed mutagenesis. Rosetta 2(DE3) cells transformed with the expression plasmid were induced with 0.5 mM IPTG at OD$_{600}$ of 0.6 for 16 h at 18 °C. Cells were harvested and lysed in cell lysis buffer (50 mM Tris-HCl pH 8.0, 300 mM NaCl, 10% glycerol, and 3 mM dithiothreitol (DTT)) by sonication. After clarification by centrifugation, glutathione agarose beads (Thermo Scientific) were added to the supernatant and incubated at 4 °C for 2 h with mixing. The beads were washed thoroughly with cell lysis buffer supplemented with 0.1% NP40. The bound GST-SUZ12 was eluted with the cell lysis buffer supplemented with 20 mM glutathione. Eluted proteins were further purified on a Superdex 200 size exclusion column (GE Healthcare) equilibrated with 20 mM Tris-HCl pH 8.0, 100 mM NaCl, and 2 mM DTT.

## Crystallization and structure determination

The truncated minimal EZH1–EED–SUZ12(VEFS) complex at 10 mg/ml was pre-incubated with 0.5 mM H3K27M peptide, 0.5 mM H3K27me3 peptide, and 1 mM SAM for 1 h on ice before crystallization. The initial crystallization conditions were screened by the sitting drop vapor diffusion method at 22 °C. Conditions obtained from the initial screens were optimized using the hanging-drop vapor diffusion method. Crystals were grown by mixing 1 µl protein solution at 10 mg/ml with 1 µl of the reservoir solution containing 10% PEG3350, 100 mM ammonium sulfate, and 50 mM HEPES pH 6.8. Diffraction-quality crystals were cryoprotected with the reservoir solution supplemented by 15% glycerol and flash-frozen in liquid nitrogen. Diffraction data were collected at a synchrotron light source and processed with HKL2000[67]. Scaled data were imported and used for molecular replacement with PDB 5WG6 as the search model[68,69]. The structure was refined by REFMAC5 and autoBUSTER, and refinement statistics were generated by PHENIX[70–72]. Model building and iterative refinement were carried out using Coot[73]. Structure figures were generated by PyMOL[74].

The crystal had a C2 space group, and two copies of complexes were found in one asymmetric unit, with one copy displaying a noticeably higher degree of mobility.

## Nucleosome reconstitution

Reconstitution of mononucleosomes was performed using the salt dialysis method. Briefly, *Xenopus laevis* histone octamers and 147-bp "601" DNA were mixed for 2 h in a buffer containing 2 M NaCl, 10 mM Tris-HCl pH 7.5, 0.1 mM EDTA, and 1 mM 2-mercaptoethanol (BME). The mixture was subjected to sequential salt dialysis in the same buffer with reduced salt concentration like this: 1 M NaCl for 2 h, 0.8 M NaCl for 2 h, 0.6 M NaCl for 2 h, 0.3 M NaCl for 2 h, 0.15 M NaCl for overnight, and 0 M NaCl for 4 h. Mononucleosomes with a tailless histone H3 lacking residue 1–27 were reconstituted following the same procedure.

## Histone methyltransferase assay

For the enzymology study, the reaction buffer contains 25 mM Tris pH 8.0, 10 mM NaCl, 1 mM EDTA, 2.5 mM MgCl$_2$ and 2.5 mM DTT. In each 20 µl reaction system, 50 nM PRC2-5m was incubated with indicated concentrations of biotin-labeled H3 (residues 21–44) peptide (Anaspec, Cat No. AS-64440), Adenosyl-L-Methionine, S-[methyl-3H]- (SAM-$^3$H) (PerkinElmer, Cat No. NET155H001MC), and SAM (NEB, Cat No. B9003S) at 30 °C for 1 h. 100 µM peptide was used for the substrate peptide-saturating condition, and 64.6 µM SAM (64 µM cold SAM plus 0.6 µM hot SAM) was used for the SAM-saturating condition. For quantification by a scintillation counter, the reaction system was stopped by adding 1 mM cold SAM. 10 µl stopped reaction mixture was then spotted onto P81 phosphocellulose paper (Reaction Biology Corporation) and air-dried for 3 h. P81 paper was washed with 50 ml of 50 mM Na$_2$CO$_3$/NaHCO$_3$ at pH 9.0 for 5 times, briefly rinsed with acetone, air-dried for 1 h, and immersed in 4 ml of scintillation fluid. The radioactive activity was quantified according to disintegrations per minute (DPM).

The reaction of the ternary complexes, EZH1–EED–SUZ12(VEFS) and EZH2–EED–SUZ12(VEFS), with nucleosomal substrates was carried out following the same protocol, except that 150 and 450 nM enzymes, 300 nM mononucleosomes, and 640 nM SAM were used in each reaction. In the case of the reaction using PRC2-5m, 50 and 100 nM enzymes were used. For quantification by autoradiography, the reaction was quenched by adding 7 µl of 4× sample loading dye and boiling at 85 °C for 5 min. The reaction mixture was separated by SDS–PAGE, followed by exposure to X-ray film to detect the methylation level.

## In vitro phosphorylation of GST-SUZ12

For the in vitro phosphorylation assay, the reaction buffer contains 50 mM Tris-HCl pH 7.5, 10 mM MgCl$_2$, 0.1 mM EDTA, 2 mM DTT, and

0.01% Brij35. In each 20 μl reaction system, 2 μg GST-SUZ12 was incubated with 100 ng CK2α2β2 or CK2α′2β2, supplemented with 200 μM ATP, at 30 °C for 30 min. The reaction was quenched by adding 7 μl of 4× sample loading dye and boiling at 85 °C for 5 min. The reaction mixture was separated by SDS–PAGE, followed by western blotting to detect the phosphorylation level.

### In vitro dephosphorylation and re-phosphorylation of PRC2 complexes

For the dephosphorylation and re-phosphorylation of the PRC2-EZH1 minimal complex, 200 μg PRC2-EZH1 was incubated with 1 μl λ phosphatase (NEB, Cat No. P0753) in 50 μl reaction buffer (50 mM HEPES 7.5, 100 mM NaCl, 2 mM DTT, 0.01% Brij35, supplemented with 1 mM MnCl$_2$) at 30 °C for 1 h, followed by the purification using Superdex 200 in the gel filtration buffer (100 mM NaCl, 20 mM Tris 8.0, and 2 mM DTT). 100 μg dephosphorylated PRC2-EZH1 was re-phosphorylated by 2 μl CK2 (NEB, Cat No. P6010) in 50 μl reaction buffer (50 mM Tris 7.5, 10 mM MgCl2, 0.1 mM EDTA, 2 mM DTT, 0.01% Brij35, supplemented with 200 μM ATP) at 30 °C for 15 min or 30 min. The dephosphorylation and re-phosphorylation efficiencies were checked by western blot. WT Sf9-expressed human PRC2-5m was treated in the same way, except that 300 μg of the complex was dephosphorylated and 150 μg of the dephosphorylated complex was re-phosphorylated.

### Mass spectrometry analysis of phosphorylated SUZ12

mESCs were harvested in ice-cold PBS containing PMSF and protease inhibitor cocktail. Pelleted cells were lysed by hypotonic buffer (10 mM HEPES pH 7.9, 1.5 mM MgCl$_2$, 10 mM KCl, 0.5% NP040, 2 mM DTT, 1 mM PMSF, 1× protease inhibitor cocktail) on ice for 30 min and centrifuged at 1000 g for 10 min to collect nuclei. The pelleted nuclei were lysed with nuclear extraction buffer (20 mM HEPES pH 7.9, 1.5 mM MgCl$_2$, 420 mM KCl, 20% glycerol, 2 mM DTT, 1 mM PMSF, 1× Protease inhibitor cocktail) by rotating at 4 °C for 1 h, followed by centrifuging at 17,000 g for 10 min. Nuclear extracts were diluted with 1 volume of hypotonic buffer and immunoprecipitated by anti-SUZ12 resins made with cyanogen bromide (CNBr)-activated Sepharose-4B (Sigma, Cat No. 9142). Captured materials were separated by SDS–PAGE and stained with Gel-Code Blue (Thermo Scientific, Cat No. 24594). The gel band containing SUZ12 was excised and submitted for MS/MS analysis.

Samples were digested overnight with trypsin (Pierce) following reduction and alkylation with DTT and iodoacetamide (Sigma–Aldrich). The samples then underwent solid-phase extraction cleanup with an Oasis HLB plate (Waters), and the resulting samples were injected onto an Orbitrap Fusion Lumos mass spectrometer coupled to an Ultimate 3000 RSLC-Nano liquid chromatography system. Samples were injected onto a 75 μm i.d., 75-cm long EasySpray column (Thermo) and eluted with a gradient from 0–28% buffer B over 90 min. Buffer A contained 2% (v/v) ACN and 0.1% formic acid in water, and buffer B contained 80% (v/v) ACN, 10% (v/v) trifluoroethanol, and 0.1% formic acid in water. The mass spectrometer operated in positive ion mode with a source voltage of 1.5 kV and an ion transfer tube temperature of 275 °C. MS scans were acquired at 120,000 resolution in the Orbitrap, and up to 10 MS/MS spectra were obtained in the ion trap for each full spectrum acquired using higher-energy collisional dissociation (HCD) for ions with charges 2–7. Dynamic exclusion was set for 25 s after an ion was selected for fragmentation.

Raw MS data files were analyzed using Proteome Discoverer v2.4 SP1 (Thermo), with peptide identification performed using Sequest HT searching against the mouse protein database from UniProt or the human protein database from UniProt with the sequence of the fusion protein EZH1-SUZ12 included. Fragment and precursor tolerances of 10 ppm and 0.6 Da were specified, and three missed cleavages were allowed. Carbamidomethylation of Cys was set as a fixed modification, with oxidation of Met and phosphorylation of Ser, Thr, and Tyr set as a variable modification. The false-discovery rate (FDR) cutoff was 1% for all peptides. Peptide abundances are defined as the peak intensity of the most abundant charge state for the peptide ion.

### Native gel shift nucleosome binding assay

0.5 nM nucleosomes were incubated with PRC2-5m (2-fold serial dilution from 2 μM) in a 20 μl reaction system (10 mM Tris 8.0, 50 mM NaCl, and 10% Glycerol) on ice for 30 min. Each 10 μl reaction mixture was separated with a 4% native polyacrylamide gel (Acrylacrylamide/Bis 60:1) in 1× TGE buffer (25 mM Tris, 190 mM Glycine, 1 mM EDTA) at 100 V for 1 h on ice. The native gel was stained by SYBR Gold. Binding assays were performed in three replicates for both WT and mutant PRC2-5m complexes, which were expressed in HEK293T cells. The gel band was quantified in ImageJ, and the dissociation constant $K_d$ was calculated by fitting binding curves in GraphPad Prism.

### Chromatin binding assay in mESCs

mESCs expressing 3×FLAG-SUZ12 (WT or S583A) were harvested with ice-cold PBS containing PMSF and 1× protease inhibitor cocktail. Pelleted cells were lysed by the hypotonic buffer on ice for 30 min and centrifuged at 1000 g for 10 min to collect nuclei. The nuclei were sonicated in binding buffer (50 mM Tris-HCl pH 8.0, 150 mM NaCl, 2 mM DTT, 10% glycerol, 0.1% NP40, 2 mM DTT, 1 mM PMSF, 1× protease inhibitor cocktail) and clarified by centrifugation at 17,000 g for 10 min. The clarified supernatant was incubated with anti-FLAG beads (Thermo Scientific, Cat No. PIA36797) at 4 °C for 1 h, followed by washing with binding buffer for three times. Captured chromatin fragments were eluted from the beads by 1.5 mg/ml FLAG peptide and analyzed by western blot to detect histone H3.

### Reporter gene repression assay

SUZ12 knockout HEK293T cells were made in the lab previously using the CRISPR/Cas9 gene-editing system[11]. The reporter vector with 6×GAL4UAS-TK-luciferase (*G6-TK-luc*) was also previously generated[58]. DNA fragments encoding GAL4DBD-HA-SUZ12 were cloned into the pCS2+ vector between its EcoRI and XhoI sites. SUZ12 knockout HEK293T cells were plated at a density of ~0.35×10$^6$ cells per well in 6-well plates and cultured for 20 h before transfection. After growing for 20 h, cells were co-transfected with 200 ng *G6-TK-luc* reporter plasmid, 200 ng pCS2+ plasmids expressing GAL4DBD-HA-SUZ12 (WT or mutant) or GAL4DBD-HA control protein, and 100 ng pCMV-β-galactosidase vector. Cells were harvested 48 h post transfection. The luciferase activity was then measured using the Luciferase Assay System kit (Promega, Cat No. E4030). Luciferase signals were normalized by β-galactosidase activity using the β-galactosidase Enzyme Assay System (Promega, Cat No. E2000). Western blot using the anti-HA antibody was performed to compare the GAL4-HA-SUZ12 expression level. GAPDH or Tubulin served as the protein loading control.

### ChIP-qPCR

mESCs were crosslinked with 1% formaldehyde for 10 min at room temperature. Formaldehyde was quenched with 0.125 M glycine, and cells were washed twice with ice-cold PBS. Cell lysates were prepared with Farnham lysis buffer (5 mM PIPES pH 8.0, 85 mM KCl, 0.5% NP40, 1 mM DTT and 1× protease inhibitor cocktail) to collect nuclei. Nuclei were resuspended with lysis buffer (50 mM Tris-HCl pH 7.9, 10 mM EDTA, 1% SDS, 1 mM DTT, and 1× protease inhibitor cocktail), and chromatin was sheared to an average size of 200–600 bp using the Covaris M220 Focused Ultrasonicator. The sheared chromatin was diluted 10-fold with ChIP dilution buffer (20 mM Tris-HCl pH 7.9, 2 mM EDTA, 150 mM NaCl, 0.5% Triton X-100, 1 mM DTT and 1× protease inhibitor cocktail). The chromatin solution was clarified by centrifugation at 15,000 g at 4 °C for 10 min. 20 μg of chromatin was used for H3K27me3-ChIP and 50 μg for FLAG ChIP. Chromatin was incubated with 5 μg of antibody overnight at 4 °C with rotation and then 80 μl of

Protein A (Invitrogen, Cat No. 10002D) or Protein G (Invitrogen, Cat No. 10004D) Dynabeads were added to the antibody-chromatin complex. After incubation at 4 °C for 2 h, beads were sequentially washed with low salt (20 mM Tris-HCl pH 8.0, 2 mM EDTA, 1% Triton X-100, 0.1% SDS, 150 mM NaCl), high salt (20 mM Tris-HCl pH 8.0, 2 mM EDTA, 1% Triton X-100, 0.1% SDS, 500 mM NaCl), LiCl (10 mM Tris-HCl pH 8.0, 1 mM EDTA, 1% NP40, 1% sodium deoxycholate, 250 mM LiCl), and TE (20 mM Tris-HCl pH 8.0, 1 mM EDTA) wash buffers. All washes were carried out at 4 °C for 10 min with rotation. The immunoprecipitated chromatin was eluted with elution buffer (1% SDS and 100 mM $NaHCO_3$). To reverse the crosslinks, samples were incubated in a 65 °C water bath for 8–12 h. RNase A and proteinase K treatment were performed before phenol:chloroform:isoamyl alcohol (25:24:1) extraction. Quantitative PCR at specific loci was performed to analyze the enrichment of FLAG-SUZ12 and H3K27me3. Primers used for qPCR are listed in Supplementary Table 3.

### EB formation and replating assay

mESCs were induced to differentiate to EBs in hanging drops. Trypsinized cells were resuspended in EB differentiation medium (DMEM, 15% FBS, 1× MEM-NEAA, 50 μM BME, 1× sodium pyruvate, 1× Pen/Strep), and 30 μl droplets of the suspension (300 cells/drop) were deposited on the lid of a 15 cm petri dish (120 drops/lid) for 48 h. Each culture plate was filled with 15 ml of 1× PBS. The EBs were then transferred to uncoated 10 cm Petri dishes and cultured on an orbital shaker at 50 rpm. EBs were harvested on day 4 and dissociated with trypsin to form single-cell suspensions. The cell suspensions were seeded in the 2i ES cell medium at a density of 30,000 cells/ml in 12-well plates and incubated for 5 days. Culture medium was changed every day. Cell colonies were stained using the Stemgent AP staining Kit II (Stemgent, Cat No. 00-0055) following the manufacturer's protocol. Experiments were performed in three replicates. Stained colonies were quantified in ImageJ, and statistics were generated in GraphPad Prism.

### Reporting summary

Further information on research design is available in the Nature Research Reporting Summary linked to this article.

## Data availability

The data that support this study are available from the corresponding author upon reasonable request. The crystal structure described in this study has been deposited in the Protein Data Bank under the accession number 7TD5. The LC-MS/MS data files have been deposited to the ProteomeXchange Consortium (http://proteomecentral.proteomexchange.org) via the MassIVE partner repository with the dataset identifier MSV000088683. Source Data are provided with this paper.

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

## Acknowledgements

The SUZ12 KO mESC line was a gift from Dr Kristian Helin from Institute of Cancer Research, London. The cDNAs of human PRC2 core components

were kindly provided by Dr Robert E. Kingston. The authors acknowledge the UTSW Proteomics Core facility for assistance with the phosphopeptide LC-MS/MS experiments. This research was supported by Welch Foundation research grant I-1790 and NIH grants GM121662 and GM 136308 to Xin L. Xin L. is a W.W. Caruth, Jr, Scholar in Biomedical Research. This research also received support from the Cecil H. and Ida Green Center Training Program in Reproductive Biology Sciences Research. L.G. was supported by American Heart Association Postdoctoral Fellowship 19POST34450043. L.J. was supported by National Natural Science Foundation of China grant 32071213. Results shown in this report are derived from work performed at Argonne National Laboratory, Structural Biology Center (SBC) at the Advanced Photon Source. SBC-CAT is operated by UChicago Argonne, LLC, for the U.S. Department of Energy, Office of Biological and Environmental Research under contract DE-AC02-06CH11357. Use of the Stanford Synchrotron Radiation Lightsource, SLAC National Accelerator Laboratory, is supported by the U.S. Department of Energy, Office of Science, Office of Basic Energy Sciences under Contract No. DE-AC02-76SF00515. The SSRL Structural Molecular Biology Program is supported by the DOE Office of Biological and Environmental Research, and by the National Institutes of Health, National Institute of General Medical Sciences (including P41GM103393). The contents of this publication are solely the responsibility of the authors and do not necessarily represent the official views of NIGMS or NIH.

## Author contributions

Xin L. conceived the study. L.G., Xiuli L., L.J., and Xin L. designed the experiments. L.G., Xiuli L., and L.J. performed the experiments with assistance from X.Y. A.L. analyzed the mass spectrometry data. L.G., Xiuli L. and Xin L. wrote the manuscript.

## Competing interests

The authors declare no competing interests.
