## [Peer Review File · Nature Communications]

REVIEWER COMMENTS

Reviewer #1 (Remarks to the Author):

Gong et al report the identification of a phosphoserine-based effect on PRC2 enzyme activity. The authors identify S583 on SUZ12 VEFS domain to be the key residue that is phosphorylated by Casein kinase 2 (CK2). The authors examine the role of S583p in PRC2 enzymatic function, chromatin binding using mono-nucleosome substrate, as well as its role in repressing genes using a luciferase reporter construct. Lastly, the authors also examine the effect of a loss of S583p in PRC2 targeting and H3K27me3 deposition during mESC differentiation.

Overall, the authors present an interesting idea on the regulation of PRC2 activity via phosphorylation of a core member of the PRC2 complex. However, there are several questions and clarifications that the authors need to address to make it clear for the reader before the paper can be accepted. Please see below

Comments:

1. The authors must include uncropped images used in all the figures in the supplement. There is no reason not to and this is becoming the norm now. The use of cropped images in main figures for illustration is ok but the original uncropped images must be included in the supplement.
2. The authors make a point that the phospho-serine antibody shows 32-fold preference over non-phosphorylated suz12 peptide containing 583 aa. While it is generally observed that phospho-antibodies are not very specific, and this is also observed in the Fig 1.d where there are faint bands detected even when S583A mutation is present. Due to this ambiguity, the author's claim about the differences in S583p in different cells (Fig 1e) is a bit skeptical. In the context of the rest of the study, I feel this is not required. It is hard to reliably quantify the differences from phospho-serine antibody based detection when the antibody is not highly specific.
3. The authors test the role of CK2 with an isolated SUZ12 or via shRNA knockdown of CK2 subunits in cells. However, In the case of the former, the authors did not use the S583p antibody instead opting to use a phospho-serine antibody which detects all serine phosphorylation. Could the authors clarify this choice? Also, the authors could demonstrate this unambiguously by using a phosphorylation reaction of PRC2 with CK2 in vitro followed by mass spec to show that S583 is the target for CK2. Human PRC2 catalytic module has been expressed and purified from E. coli using a construct very similar to the PRC2-EZH1 construct used here (Brooun et al. 2016). This would allow for the authors to test for the site of

phosphorylation in a more unambiguous setting. In my opinion, this would be far more concrete than the assay currently presented and would unambiguously state that CK2 is the kinase for S583p. The authors should also use the S583p antibody developed rather than a generic phospho-serine antibody.

4. Fig 2c,d bring up the same issue with respect to the specificity of phospho-serine antibody and how relevant the reduction seen here are compared to non-specific reduction in serine phosphorylation in cells due to CK2 shRNA based knockdown. Fig 2c,d would have been ok if the authors demonstrated more convincingly in vitro that S583 was the target site for CK2 (see above # 3)

5. Pg. 10 Type (VESF -> VEFS)

6. The authors choice of PRC2-EZH1 catalytic module for structure studies and comparison is also unclear. While it is understandable from a crystallization requirement (which has also included the replacement of several segments of EZH2 with GSGS-like linker segments), previous studies have shown that PRC2-EZH1 shows poor methyltransferase activity compared to PRC2-EZH2. It is unclear how the enzyme kinetics results (Fig 4) translate to PRC2-EZH2 and therefore remain ambiguous. For figure 4c, The authors also don't state the concentration of SAM and H3 peptide used in H3-saturation conditions and SAM saturation conditions, respectively.

7. In Pg. 10 under S583 phosphorylation stabilizes enzyme active site heading, the authors state that acidic loop containing S583 is not well ordered in known structures in the absence of S583 phosphorylation. While 5HYN represents a structure of the PRC2-EZH2 catalytic module isolated from bacteria (one expects S583 to be non-phosphorylated), 5IJ7 represents PRC2-EZH2 isolated from insect cells where S583 is atleast partially phosphorylated. Moreover comparison of 5HYN, 5IJ7 with cryo-EM structures (6C24 and 6WKR) suggests that PRC2 isolated from insect cells all likely retain some phosphorylation (presumably?) and exist in a state which seems similar to the crystal structure presented here. The authors use this potential difference between a S583 non-phosphorylated (represented by a state similar to 5HYN) versus one that is phosphorylated (5IJ7, 6C24, 6WKR) to make the case that phosphorylation results in interaction with K684 of EZH2. However, the single turn helix of EZH2 (SET) containing K684 is in a very similar conformation in all currently existing PRC2 structures including 5HYN, 5IJ7, 6C24, 6WKR and hence it is unclear how S583 phosphorylation results in a change via K684 when the latter's conformation seems independent of S583 phosphorylation status.

8. The effect of S583 phosphorylation using mutations in EZH1 suffer from similar concerns as in #6 where EZH1-PRC2 inherently shows much lower methyltransferase activity compared to EZH2-PRC2. It would be better if the authors showed activity assays with PRC2-EZH2 catalytic module to solidify their claim about the effect of S583 phosphorylation and enzymatic activity of PRC2.

9. The authors also claim nucleosome binding defect in Fig 4a using K684A mutation or SUZ12 mutations 567A and 568A. The authors show instead directly show the nucleosome binding effect of S583A and this perhaps should be done within the PRC2-EZH2 catalytic module context as EZH1 has previously shown to independently confer nucleosome binding activity in PRC2-EZH1. This would help authors avoid having to take into account the inherent EZH1- contributed nucleosome binding versus the effect of S583A mutation.

10. It is quite hard for the reader to judge the changes in enzyme activity from Fig 4b as the authors just state histone methylation and do not make it clear whether this is detecting defects in H3K27me or H3K27me2 or H3K27me3. Could the authors clarify this? Could the authors verify that the PRC2-5m isolated from HEK cells carry no S583 phosphorylation even in a PRC2-5m S583A mutant complex. Another clarification that authors could provide that would help gauge this better is whether it is possible that the the isolated complex from HEK cells have wild type endogenous SUZ12 that could have been partially incorporated into this complex?

11. For the nucleosome binding assays, the assay that the authors show test for the presence of EZH2 and not binding. The inherent assumption being presence of EZH2 is indicative of binding. It would have been far better if the authors carry out native gel shift binding assays or something similar to show that PRC2-5m harboring S583A or S583p affects nucleosome binding. It is hard for the reader to interpret binding versus EZH2 presence as presented by the authors.

12. It would have also been informative if the authors presented S583D or some other phosphomimic to make the case for phosphorylation or the negative charge-based interaction network being important for enzyme activity rather than or in addition to the S583A. One of the structures (5IJ7) the authors used for comparison indeed uses S583D in their structural study which show the S583D in a similar interaction network as seen in 6C24, 6WKR as well as in the author's structure presented here. The GAL4 luciferase assays would also benefit from S583D or phosphomimic.

13. It is hard to gauge how the authors conclude slightly weaker bulk chromatin association from Fig. 6c based on the data as presented. This remains ambiguous.

In summary, the phosphorylation-based regulation of PRC2 enzymatic activity though known from earlier studies for EZH2 is less well established for SUZ12. The authors hence present an interesting idea incorporating structural and biochemical studies. However, several aspects need to be addressed and/or clarified for the study to clearly outline the effect of S583p-mediated effect on PRC2 enzymatic activity. While the reviewer understands some or many of the suggestions require new experiments, I feel it would improve the quality and scope of the manuscript as the author's idea is interesting.

Reviewer #2 (Remarks to the Author):

PRC2 plays a critical role in development and cancer. Understanding how this molecular machine is regulated is therefore important for basic research and to inspire new therapeutic strategies. Gong et al. report the structural and functional consequences of phosphorylation of SUZ12-Ser583, a component of PRC2. The authors confirm the existence of this mark and identify CK2 as the kinase responsible for its installation. Structural studies revealed a conformational change caused by phosphorylation. Residues involved in pSer binding were found to contribute to PRC2 activity and a SUZ12-S583A mutation increased the K_m for SAM. This mutation also diminished PRC2-mediated reporter gene repression, chromatin binding and cellular memory during differentiation. Collectively, this study represents a multi-pronged approach to establish a new regulatory element in this complex circuit. Accordingly, the work contributes to our understanding of an important enzyme complex and is of interest to the broad readership of Nature Comms. The experiments are well documented and support the conclusions. I therefore believe it is suitable for publication once the following issue is addressed:

The S583A mutation exhibits a strong phenotype in HMT activity and nucleosome binding assays, much more so than in the kinetic assays with peptide substrates. To exclude the possibility that the mutation affects the composition of PRC2 in unanticipated ways, the authors should repeat the activity and nucleosome binding assays (Figure 4b,d) with phosphatase-treated PRC2 containing wild-type SUZ12.

Minor corrections:

- In Figure 3d, basic residues should be shown in their protonated form for consistency with their anionic counterparts; the backbone amide of K684 should be NH; H567 is shown in the wrong stereochemistry
- Page 13: The Michaelis constant (written as K_m) is not directly a measure of “binding affinity”, please rephrase. In Figure 4c, the rate constant should be written as k_{cat}

Reviewer #3 (Remarks to the Author):

Dear Xin Liu,

Overall, I would endorse your manuscript CK2-Mediated Phosphorylation of SUZ12 Promotes PRC2 Function by Stabilizing Enzyme Active Site acceptance with minor revisions. This expands the knowledge of the function of PRC2 on a mechanistic level. The work supporting that CK2 is responsible for the phosphorylation of S583 by Ck2 is solid. The work in creating a phospho-specific antibody to demonstrate that this phosphorylation event occurs in cell lines is solid. Something that would strengthen the translatability of this work would be showing that this modification is present in primary breast cancer or glioma. You should clarify in the text of the abstract which species the work was performed in where you reference SUZ12 and CK2. This work is significant to the field as it contributes both biochemical and novel structural information to the roll of PRC2 enzymatic activity and biological function. Experimental support of the model is solid. One additional change I would like to see, is that the observed differences between the images of the phosphorylation mutant and the WT Suz12 in the stem cell AP Assay should be properly quantified instead of only described in the text.

The work will update the current PRC2 working model so that it is more mechanistic. Also significant to the field in general that phosphorylation can cause a physical change in a non-active site that could lead to a change of enzyme activity. The experimental work is solid, and no additional experiments are required, although additional staining of primary cancer samples to show presence of this modification (or an alternative method like phospho-Mass spec) would greatly strengthen the biology around this finding. The methodology is sound and meets the standards in the field. Enough information is provided in the paper and in the supplemental to ensure that the results could be replicated.

Reviewer #1 (Remarks to the Author):

Gong et al report the identification of a phosphoserine-based effect on PRC2 enzyme activity. The authors identify S583 on SUZ12 VEFS domain to be the key residue that is phosphorylated by Casein kinase 2 (CK2). The authors examine the role of S583p in PRC2 enzymatic function, chromatin binding using mono-nucleosome substrate, as well as its role in repressing genes using a luciferase reporter construct. Lastly, the authors also examine the effect of a loss of S583p in PRC2 targeting and H3K27me3 deposition during mESC differentiation.

Overall, the authors present an interesting idea on the regulation of PRC2 activity via phosphorylation of a core member of the PRC2 complex. However, there are several questions and clarifications that the authors need to address to make it clear for the reader before the paper can be accepted. Please see below

Comments:

1. The authors must include uncropped images used in all the figures in the supplement. There is no reason not to and this is becoming the norm now. The use of cropped images in main figures for illustration is ok but the original uncropped images must be included in the supplement.

The uncropped images are now provided in **Fig. S16, S17, S18, and S19**, which are labeled based on the main figures and are referred to in the figure legends of the main figures.

2. The authors make a point that the 1osphor-serine antibody shows 32-fold preference over non-phosphorylated suz12 peptide containing 583 aa. While it is generally observed that phosphor-antibodies are not very specific, and this is also observed in the Fig 1.d where there are faint bands detected even when S583A mutation is present. Due to this ambiguity, the author's claim about the differences in S583p in different cells (Fig 1e) is a bit skeptical. In the context of the rest of the study, I feel this is not required. It is hard to reliably quantify the differences from 1osphor-serine antibody based detection when the antibody is not highly specific.

We acknowledge the reviewer's point that like many other anti-phosphor antibodies the newly developed anti-SUZ12S583p antibody displays a degree of nonspecific binding. With that said, **Fig. 1c, 1d, and S2** together also indicated that the observed Western blot signals were largely dominated by the specific binding of the antibody to the phosphorylated SUZ12S583. Accordingly, we believe this antibody offers sufficient specificity to provide a semi-quantitative view of the phosphorylation level of SUZ12S583 in various cell lines. We removed the quantification table from the figure and modified the text to highlight the semi-quantitative nature of the Western blot on **Page 8**: *"Using the newly developed antibody, we examined SUZ12S583 phosphorylation in various cancer cell lines in a **semi-quantitative***

manner. We found that SUZ12S583 phosphorylation is a widespread phenomenon (Fig. 1e and S3). Compared to the total cellular SUZ12, the SUZ12S583 phosphorylation level displayed cell line-specific variations, with some cell lines showing distinctly less phosphorylation (Fig. 1e and S3), which may be accounted for by different kinase activities accessible to SUZ12 in these cells.”

3. The authors test the role of CK2 with an isolated SUZ12 or via shRNA knockdown of CK2 subunits in cells. However, In the case of the former, the authors did not use the S583p antibody instead opting to use a phosphor-serine antibody which detects all serine phosphorylation. Could the authors clarify this choice? Also, the authors could demonstrate this unambiguously by using a phosphorylation reaction of PRC2 with CK2 *in vitro* followed by mass spec to show that S583 is the target for CK2. Human PRC2 catalytic module has been expressed and purified from *E. coli* using a construct very similar to the PRC2-EZH1 construct used here (Brooun et al. 2016). This would allow for the authors to test for the site of phosphorylation in a more unambiguous setting. In my opinion, this would be far more concrete than the assay currently presented and would unambiguously state that CK2 is the kinase for S583p. The authors should also use the S583p antibody developed rather than a generic 2phosphor-serine antibody.

*When we performed the *in vitro* phosphorylation assay using bacterially expressed isolated SUZ12, we compared S583 and other non-S583 serine residues for the contribution to the total phosphoserine signals conferred by CK2, by surveying the individual alanine mutations. The phosphoserine signals were mostly lost for the S583A mutant, whereas the signals were only minimally affected by the alanine mutations of non-S583 residues, suggesting a specific CK2 kinase activity towards S583. Since the newly developed anti-SUZ12S583p antibody is not able to detect the phosphorylation at other serine residues, we chose to use a phosphoserine antibody that detects all serine phosphorylation. In this way, the phosphorylation level of all serine residues could be detected and compared between WT and mutant SUZ12 proteins.*

*As suggested by the reviewer, we performed the CK2-mediated *in vitro* phosphorylation assay on the human PRC2-EZH1 catalytic module. We were unable to purify the PRC2-EZH2 catalytic module from *E. coli* by following the published protocol and therefore used an alternative approach. We first used lambda phosphatase to erase the existing SUZ12SS583p in PRC2-EZH1 expressed in yeast (Fig. S6a). The phosphatase was then removed by size exclusion chromatography. We next added human CK2 to the dephosphorylated PRC2-EZH1, and the kinase reaction mixture was subjected to mass spectrometry analysis (Fig. S6b). The new results indicated that only S583 but no other captured serine residues were significantly phosphorylated by human CK2. Texts were revised on Page 11: “In addition, human CK2 was able to specifically phosphorylate S583 within the truncated PRC2-EZH1 minimal complex pretreated by λ protein phosphatase, confirming the CK2 kinase specificity in this context (Fig. S6).”*

4. Fig 2c,d bring up the same issue with respect to the specificity of phospho-serine antibody and how relevant the reduction seen here are compared to non-specific reduction

in serine phosphorylation in cells due to CK2 shRNA based knockdown. Fig 2c,d would have been ok if the authors demonstrated more convincingly in vitro that S583 was the target site for CK2 (see above # 3)

See the response to #3 above. Results in the revision have now confirmed that S583 is the major target site for CK2 *in vitro*.

5. Pg. 10 Type (VESF -> VEFS)

Corrected.

6. The authors choice of PRC2-EZH1 catalytic module for structure studies and comparison is also unclear. While it is understandable from a crystallization requirement (which has also included the replacement of several segments of EZH2 with GSGS-like linker segments), previous studies have shown that PRC2-EZH1 shows poor methyltransferase activity compared to PRC2-EZH2. It is unclear how the enzyme kinetics results (Fig 4) translate to PRC2-EZH2 and therefore remain ambiguous. For figure 4c, The authors also don't state the concentration of SAM and H3 peptide used in H3-saturation conditions and SAM saturation conditions, respectively.

We actually used EZH2-containing PRC2-5m (EZH2–EED–SUZ12–RBBP4–AEBP2) for the enzymatic assay in **Fig. 4c** (original Fig. 4b) and the enzymology study in **Fig. 4e** (original Fig. 4c). Moreover, in response to #8 below, we performed additional enzymatic assays using the PRC2-EZH2 catalytic module and included the results in **Fig. 4b** in the revision. Texts were revised on **Page 13**: “*Similar results were obtained for the same set of mutations in the context of the EZH2-containing minimal PRC2 complex (Fig. 4b).*”

SAM and H3 peptide concentrations were provided in the Methods section in the original submission and are also noted in **Fig. 4e** in the revision now.

7. In Pg. 10 under S583 phosphorylation stabilizes enzyme active site heading, the authors state that acidic loop containing S583 is not well ordered in known structures in the absence of S583 phosphorylation. While 5HYN represents a structure of the PRC2-EZH2 catalytic module isolated from bacteria (one expects S583 to be non-phosphorylated), 5IJ7 represents PRC2-EZH2 isolated from insect cells where S583 is at least partially phosphorylated. Moreover comparison of 5HYN, 5IJ7 with cryo-EM structures (6C24 and 6WKR) suggests that PRC2 isolated from insect cells all likely retain some phosphorylation (presumably?) and exist in a state which seems similar to the crystal structure presented here. The authors use this potential difference between a S583 non-phosphorylated (represented by a state similar to 5HYN) versus one that is phosphorylated (5IJ7, 6C24, 6WKR) to make the case that phosphorylation results in interaction with K684 of EZH2. However, the single turn helix of EZH2 (SET) containing K684 is in a very similar conformation in all currently existing PRC2

structures including 5HYN, 5IJ7, 6C24, 6WKR and hence it is unclear how S583 phosphorylation results in a change via K684 when the latter's conformation seems independent of S583 phosphorylation status.

The phosphoserine-centered interaction network is lacking in all the previous structure models, and the structure of the PDS loop of SUZ12 captured in the current study is apparently different from those previous models as well. We think at least two possibilities as proposed below can account for the observed structural difference. (1) S583 is not phosphorylated in 5HYN, and S583 is mutated to aspartate in 5IJ7, which may not fully mimic a phosphoserine and result in a dislodged PDS loop. (2) S583 may be partially phosphorylated in 6C24 and 6WKR, however, the local resolution of the cryo-EM maps may not be high enough to allow the structural model of the phosphoserine-centered interaction network to be built unambiguously.

In terms of the conformation of EZH1K684/EZH2K683, the saturating concentrations of the SAM/SAH and histone substrate used in the structural studies may have stabilized the conformation of this lysine residue, even in the absence of SUZ12S583 phosphorylation. In addition, the static crystal or cryo-EM structures cannot really reflect protein residue dynamics in solution. Our proposal that EZH1K684/EZH2K683 may be stabilized by the phosphoserine-centered interaction network is supported by several experimental observations below. (1) EZH1K684 is directly bound by the phosphate group in the current structure (**Fig. 3c**). (2) Alanine mutation of EZH2K683 compromised PRC2 enzymatic activity in solution, under a condition in which neither SAM nor histone substrate is saturating (**Fig. 4c**). (3) EZH1K684/EZH2K683 is a part of the SAM binding pocket based on the current and previous structures, and our enzymology study showed that the major defect caused by the loss of the SUZ12S583 phosphorylation is SAM binding by PRC2 (**Fig. 4e**).

8. The effect of S583 phosphorylation using mutations in EZH1 suffer from similar concerns as in #6 where EZH1-PRC2 inherently shows much lower methyltransferase activity compared to EZH2-PRC2. It would be better if the authors showed activity assays with PRC2-EZH2 catalytic module to solidify their claim about the effect of S583 phosphorylation and enzymatic activity of PRC2.

In the revision, we showed additional enzymatic assays using the PRC2-EZH2 catalytic module and included the results in **Fig. 4b**. EZH2-containing PRC2-5m (EZH2-EED-SUZ12-RBBP4-AEBP2) was used for the enzymatic assay in **Fig. 4c** and the enzymology study in **Fig. 4e**.

9. The authors also claim nucleosome binding defect in Fig 4a using K684A mutation or SUZ12 mutations 567A and 568A. The authors show instead directly show the nucleosome binding effect of S583A and this perhaps should be done within the PRC2-EZH2 catalytic module context as EZH1 has previously shown to independently confer nucleosome binding activity in PRC2-EZH1. This would help authors avoid having to take into account the inherent EZH1- contributed nucleosome binding versus the effect of S583A mutation.

To address this reviewer's concern and in connection to #11 below, we performed native gel shift nucleosome binding assays using EZH2-containing PRC2-5m (EZH2–EED–SUZ12–RBBP4–AEBP2) WT and SUZ12S583A mutant complexes. The mutant complex displayed a 2-fold reduction in the nucleosome binding affinity. The new results were included in **Fig. 4h** and **Fig. S12** in the revision. Texts were revised on **Page 15**: *“To gain a quantitative view of nucleosome binding, we performed native gel shift assays. The nucleosome binding affinity of the S583A mutant PRC2-5m complex was reduced by roughly two folds compared to that of the WT complex (Fig. 4h and S12).”*

10. It is quite hard for the reader to judge the changes in enzyme activity from Fig 4b as the authors just state histone methylation and do not make it clear whether this is detecting defects in H3K27me or H3K27me2 or H3K27me3. Could the authors clarify this? Could the authors verify that the PRC2-5m isolated from HEK cells carry no S583 phosphorylation even in a PRC2-5m S583A mutant complex. Another clarification that authors could provide that would help gauge this better is whether it is possible that the the isolated complex from HEK cells have wild type endogenous SUZ12 that could have been partially incorporated into this complex?

The enzymatic assays presented in **Fig. 4c** (original Fig. 4b) were SAM-³H-based radioactive assays, which cannot distinguish the methylation state of the product. In the revision, we used Western blot-based enzymatic assays to study the effect of the SUZ12S583A mutation on the methylation multiplicity and showed that all three methylation states were comparably impaired (**Fig. S11**). Texts were revised on **Page 13**: *“When SUZ12 harbors the S583A single mutation and thus lacks phosphorylation at this site, histone methylation was severely compromised (Fig. 4c). All methylation states were affected by the S583A mutation (Fig. S11).”*

Components of PRC2-5m were all overexpressed under a strong CMV promoter in HEK293T cells. The amount of endogenous SUZ12 is likely negligible in this case. The endogenous phosphorylated SUZ12 is not detected in the purified mutant complex, at least within the detection limit of the anti-SUZ12S593p antibody (**Fig. S10**). The text was revised on **Page 13**: *“No endogenous phosphorylated SUZ12 was detected in the purified SUZ12S583A mutant complex (Fig. S10).”*

11. For the nucleosome binding assays, the assay that the authors show test for the presence of EZH2 and not binding. The inherent assumption being presence of EZH2 is indicative of binding. It would have been far better if the authors carry out native gel shift binding assays or something similar to show that PRC2-5m harboring S583A or S583p affects nucleosome binding. It is hard for the reader to interpret binding versus EZH2 presence as presented by the authors.

As suggested, results for native gel shift nucleosome binding assays are now provided in **Fig. 4h** and **Fig. S12** in the revision. The SUZ12S583A mutation reduced the nucleosome binding affinity of EZH2-containing PRC2-5m by two folds. Texts were revised on **Page 15**:

“To gain a quantitative view of nucleosome binding, we performed native gel shift assays. The nucleosome binding affinity of the S583A mutant PRC2-5m complex was reduced by roughly two folds compared to that of the WT counterpart (Fig. 4h and S12).”

12. It would have also been informative if the authors presented S583D or some other phosphomimetic to make the case for phosphorylation or the negative charge-based interaction network being important for enzyme activity rather than or in addition to the S583A. One of the structures (5IJ7) the authors used for comparison indeed uses S583D in their structural study which show the S583D in a similar interaction network as seen in 6C24, 6WKR as well as in the author’s structure presented here. The GAL4 luciferase assays would also benefit from S583D or phosphomimetic.

We included the SUZ12S583D phosphomimetic mutation in the enzymatic assays shown in Fig. 4c in the revision. This mutation largely rescued the enzymatic activity of the SUZ12S583A mutant. The text was revised on Page 13: “In comparison, the S583D phosphomimetic mutant complex did not display a defect in catalysis (Fig. 4c).”

We also included the SUZ12S583D phosphomimetic mutation in the reporter gene assays. As shown in Fig. S14, this mutation largely restored the gene repression activity of the S583A mutant PRC2, to a level comparable to that of WT PRC2. The text was revised on Page 16: “In support of the role of S583 phosphorylation, the S583D phosphomimetic mutation was not found to compromise the reporter gene repression (Fig. S14).”

13. It is hard to gauge how the authors conclude slightly weaker bulk chromatin association from Fig. 6c based on the data as presented. This remains ambiguous.

*Anti-SUZ12 antibody pulled down a fraction of fragmented chromatin as indicated by the presence of histone H3 from the WT mESCs, whereas the same antibody did not capture any H3 from the SUZ12S583A mutant mESCs. We explained this experimental result by proposing that the lack of SUZ12S583 phosphorylation in the latter case caused defective chromatin binding. This explanation is supported by the *in vitro* nucleosome binding assays shown in the previous sections (Fig. 4f and 4h) as well as the ChIP results shown in the next section (Fig. 6d).*

In summary, the phosphorylation-based regulation of PRC2 enzymatic activity though known from earlier studies for EZH2 is less well established for SUZ12. The authors hence present an interesting idea incorporating structural and biochemical studies. However, several aspects need to be addressed and/or clarified for the study to clearly outline the effect of S583p-mediated effect on PRC2 enzymatic activity. While the reviewer understands some or many of the suggestions require new experiments, I feel it would improve the quality and scope of the manuscript as the author's idea is interesting.

Reviewer #2 (Remarks to the Author):

PRC2 plays a critical role in development and cancer. Understanding how this molecular machine is regulated is therefore important for basic research and to inspire new therapeutic strategies. Gong et al. report the structural and functional consequences of phosphorylation of SUZ12-Ser583, a component of PRC2. The authors confirm the existence of this mark and identify CK2 as the kinase responsible for its installation. Structural studies revealed a conformational change caused by phosphorylation. Residues involved in pSer binding were found to contribute to PRC2 activity and a SUZ12-S583A mutation increased the K_m for SAM. This mutation also diminished PRC2-mediated reporter gene repression, chromatin binding and cellular memory during differentiation. Collectively, this study represents a multi-pronged approach to establish a new regulatory element in this complex circuit. Accordingly, the work contributes to our understanding of an important enzyme complex and is of interest to the broad readership of Nature Comms. The experiments are well documented and support the conclusions. I therefore believe it is suitable for publication once the following issue is addressed:

The S583A mutation exhibits a strong phenotype in HMT activity and nucleosome binding assays, much more so than in the kinetic assays with peptide substrates. To exclude the possibility that the mutation affects the composition of PRC2 in unanticipated ways, the authors should repeat the activity and nucleosome binding assays (Figure 4b,d) with phosphatase-treated PRC2 containing wild-type SUZ12.

As suggested, we repeated the activity and nucleosome binding assays using phosphatase-treated and CK2-rephosphorylated PRC2. Results are now provided in **Fig. 4d** and **Fig. 4g**. The text is revised on **Page 14**: “*More directly, CK2-mediated in vitro phosphorylation of λ phosphatase-treated WT PRC2-5m pronouncedly enhanced histone methylation (Fig. 4d)*”, and on **Page 15**: “*Congruently, nucleosomes were bound less tightly by λ phosphatase-treated WT PRC2-5m, compared to the same complex re-phosphorylated by CK2 in vitro (Fig. 4g).*”

Minor corrections:

- In Figure 3d, basic residues should be shown in their protonated form for consistency with their anionic counterparts; the backbone amide of K684 should be NH; H567 is shown in the wrong stereochemistry

Corrected.

- Page 13: The Michaelis constant (written as K_m) is not directly a measure of “binding affinity”, please rephrase. In Figure 4c, the rate constant should be written as k_{cat}

On **Page 14**, the sentence was rephrased to: “*As indicated by the K_m values changing from*

0.5 μ M to 2.9 μ M, loss of S583 phosphorylation most profoundly affected SAM binding to PRC2, ...”.

The rate constant was corrected to k_{cat} in **Fig. 4e** (original Fig. 4c) in the revision.

Reviewer #3 (Remarks to the Author):

Dear Xin Liu,

Overall, I would endorse your manuscript CK2-Mediated Phosphorylation of SUZ12 Promotes PRC2 Function by Stabilizing Enzyme Active Site acceptance with minor revisions. This expands the knowledge of the function of PRC2 on a mechanistic level. The work supporting that CK2 is responsible for the phosphorylation of S583 by Ck2 is solid. The work in creating a phospho-specific antibody to demonstrate that this phosphorylation event occurs in cell lines is solid. Something that would strengthen the translatability of this work would be showing that this modification is present in primary breast cancer or glioma. You should clarify in the text of the abstract which species the work was performed in where you reference SUZ12 and CK2. This work is significant to the field as it contributes both biochemical and novel structural information to the role of PRC2 enzymatic activity and biological function. Experimental support of the model is solid. One additional change I would like to see, is that the observed differences between the images of the phosphorylation mutant and the WT Suz12 in the stem cell AP Assay should be properly quantified instead of only described in the text.

As suggested by the reviewer, AP assay was quantified, and the result is now provided in Fig. 6i.

The work will update the current PRC2 working model so that it is more mechanistic. Also significant to the field in general that phosphorylation can cause a physical change in a non-active site that could lead to a change of enzyme activity. The experimental work is solid, and no additional experiments are required, although additional staining of primary cancer samples to show presence of this modification (or an alternative method like phospho-Mass spec) would greatly strengthen the biology around this finding. The methodology is sound and meets the standards in the field. Enough information is provided in the paper and in the supplemental to ensure that the results could be replicated.

In the revision, we provided additional Western blot data on the level of SUZ12S583 phosphorylation in BT-474 primary ductal carcinoma cells and MCF10A mammary gland epithelial cells (Fig. S3).

REVIEWERS' COMMENTS

Reviewer #1 (Remarks to the Author):

Gong et al report the identification of a phosphoserine-based effect on PRC2 enzyme

activity. The authors identify S583 on SUZ12 VEFS domain to be the key residue that is phosphorylated by Casein kinase 2 (CK2). The authors examine the role of S583p in PRC2 enzymatic function, chromatin binding using mono-nucleosome substrate, as well as its role in repressing genes using a luciferase reporter construct. Lastly, the authors also examine the effect of a loss of S583p in PRC2 targeting and H3K27me3 deposition during mESC differentiation.

Overall, the authors address most of the clarifications and concerns raised previously and as such the manuscript is much more clear and concise. The revised manuscript is very nice! The addition of PRC2-EZH2 assays and nucleosome binding assays were very helpful. The author's reporting of the raw images are also much appreciated. There are some minor things that need addressing (see below).

1. It would be helpful if the authors included a brief summary of the details of the search in phospho database that helped them identify CK2 as the candidate.
2. Remove line 206/207 – It appears that the authors are trying to make the case that S546 could also be a substrate for CK2 but since this is not elaborated or tested further and is not the focus of the paper, I would remove this line. The authors also make the case for no other endogenous phosphorylation being detected in a S583A background so in this context Line 206/207 about S546 is distracting from the main focus of the paper.
3. In Sig. S11, It is unclear why 50 nM PRC2 shows much higher activity than 100 nM PRC2 for both S583A and WT. This assay is with mono-nucleosome substrate (?) since it follows Fig. 4c but it is not clear why there is a drastic difference with higher conc of PRC2? Is this under SAM limiting conditions? Could the authors clarify this in the text/methods?
4. It would have been much more helpful to have the PRC2-EZH2 nucleosome binding assays for H3ΔN also shown with Native EMSAs in order to support the author's conclusion that S583A or lack of phosphorylation of S583 affects SAM binding which in turn has a direct effect on H3 tail capture by the substrate. I would suggest the authors instead edit the text [Line 315: "suggesting S583 phosphorylation "may be" necessary for optimal binding of H3 substrate tail at the enzyme active site in the context of nucleosome....]

5. Citation #64 which is a cryo-EM structure of PRC2 with H2AK119ub containing nucleosome (?) is a more direct evidence of PRC2 recruitment to chromatin so it has to be included in line 132 which instead cite studies which are mostly indirect indicators.

6. Line 84... one such study that looked at linker length DNA between nucleosome and allosteric PRC2 activation is the elegant di-nucleosome bound PRC2 from the Nogales group. This should be referenced here.

Reviewer #2 (Remarks to the Author):

The authors' new experiments and edits address all the reviewers concerns and I believe the manuscript is now suitable for publication. The only very minor point I have is to please add figure legends to the new supporting Figures S16-S19 which convey the purpose of the figures and explain the presence of additional bands in some of the full-sized blots (e.g. S16b&d; S17a,b; S19a,b).

Reviewer #1 (Remarks to the Author):

Gong et al report the identification of a phosphoserine-based effect on PRC2 enzyme activity. The authors identify S583 on SUZ12 VEFS domain to be the key residue that is phosphorylated by Casein kinase 2 (CK2). The authors examine the role of S583p in PRC2 enzymatic function, chromatin binding using mono-nucleosome substrate, as well as its role in repressing genes using a luciferase reporter construct. Lastly, the authors also examine the effect of a loss of S583p in PRC2 targeting and H3K27me3 deposition during mESC differentiation.

Overall, the authors address most of the clarifications and concerns raised previously and as such the manuscript is much more clear and concise. The revised manuscript is very nice! The addition of PRC2-EZH2 assays and nucleosome binding assays were very helpful. The author's reporting of the raw images are also much appreciated. There are some minor things that need addressing (see below).

1. It would be helpful if the authors included a brief summary of the details of the search in phospho database that helped them identify CK2 as the candidate.

More details were provided in the figure legend of Fig. 2a.

2. Remove line 206/207 – It appears that the authors are trying to make the case that S546 could also be a substrate for CK2 but since this is not elaborated or tested further and is not the focus of the paper, I would remove this line. The authors also make the case for no other endogenous phosphorylation being detected in a S583A background so in this context Line206/207 about S546 is distracting from the main focus of the paper.

This sentence was removed.

3. In Sig. S11, It is unclear why 50 nM PRC2 shows much higher activity than 100 nM PRC2 for both S583A and WT. This assay is with mono-nucleosome substrate (?) since it follows Fig. 4c but it is not clear why there is a drastic difference with higher conc of PRC2? Is this under SAM limiting conditions? Could the authors clarify this in the text/methods?

The labels for the two enzyme concentrations were accidentally switched and were corrected. Information regarding substrate and SAM concentration was provided in the figure legend.

4. It would have been much more helpful to have the PRC2-EZH2 nucleosome binding assays for H3ΔN also shown with Native EMSAs in order to support the author's conclusion that S538A or lack of phosphorylation of S583 affects SAM binding which in turn has a direct effect on H3 tail capture by the substrate. I would suggest the authors instead edit the text [Line 315: "suggesting S583 phosphorylation "may be" necessary for optimal binding of H3 substrate tail at the enzyme active site in the context of nucleosome....]

The additional assay was performed, and the result was provided in Supplementary **Fig. 12c**. The text was changed as suggested.

5. Citation #64 which is a cryo-EM structure of PRC2 with H2AK119ub containing nucleosome (?) is a more direct evidence of PRC2 recruitment to chromatin so it has to be included in line 132 which instead cite studies which are mostly indirect indicators.

The reference was cited

6. Line 84... one such study that looked at linker length DNA between nucleosome and allosteric PRC2 activation is the elegant di-nucleosome bound PRC2 from the Nogales group. This should be referenced here.

The reference was not cited to avoid confusion. The new reference suggested by the reviewer deals with the EED-mediated allosteric activation mechanism of PRC2. However, the activation mechanism mentioned in this sentence was specifically shown to be different from the EED-mediated allosteric activation mechanism in the cited reference #16 by Yuan et al.

Reviewer #2 (Remarks to the Author):

The authors' new experiments and edits address all the reviewers concerns and I believe the manuscript is now suitable for publication. The only very minor point I have is to please add figure legends to the new supporting Figures S16-S19 which convey the purpose of the figures and explain the presence of additional bands in some of the full-sized blots (e.g. S16b&d; S17a,b; S19a,b).

Figure legends and clarifications were provided.